# Frictional slip weakening and shear-enhanced crystallinity in simulated coal fault gouges at slow slip rates

Caiyuan Fan[1], Jinfeng Liu[123], Luuk B. Hunfeld[4], Christopher J. Spiers[4]

[1]School of Earth Sciences and Engineering, Sun Yat-Sen University, Guangzhou, 510275, China
[2]Guangdong Provincial Key Lab of Geodynamics and Geohazards, Sun Yat-Sen University, Zhuhai, 519082, China
[3]Southern Marine Science and Engineering Guangdong Laboratory, Zhuhai, 519082, China
[4]Department of Earth Sciences, Utrecht University, Utrecht, 3584 CB, The Netherlands

*Correspondence to*: Jinfeng Liu (liujinf5@mail.sysu.edu.cn)

**Abstract.** Previous studies show that organic-rich fault patches may play an important role in promoting unstable fault slip. However, the frictional properties of rock materials with near 100% organic content, e.g. coal, and the controlling microscale mechanisms, remain unclear. Here, we report seven velocity stepping (VS) and one slide-hold-slide (SHS) friction experiments performed on simulated fault gouges prepared from bituminous coal, collected from the upper Silesian Basin of Poland. These experiments were performed at 25–45 MPa effective normal stress and 100 ℃, employing sliding velocities of 0.1–100 μm s$^{-1}$, using a conventional triaxial apparatus plus direct shear assembly. All samples showed marked slip weakening behaviour at shear displacements beyond ~1–2 mm, from a peak friction coefficient approaching ~0.5 to (near) steady-state values of ~0.3, regardless of effective normal stress or whether vacuum dry / flooded with distilled (DI) water at 15 MPa pore fluid pressure. Analysis of both unsheared and sheared samples by means of microstructural observation, micro-area X-ray diffraction (XRD) and Raman spectroscopy suggests that the marked slip weakening behaviour can be attributed to the development of R-, B- and Y- shear bands, with internal shear-enhanced coal crystallinity development. The SHS experiment performed showed a transient peak healing (restrengthening) effect that increased with the logarithm of hold time at a linearized rate of ~0.006. We also determined the rate-dependence of steady-state friction for all VS samples using a full rate and state friction approach. This showed a transition from velocity strengthening to velocity weakening at slip velocities >1 μm s$^{-1}$ in the coal sample under vacuum dry conditions, but at >10 μm s$^{-1}$ in coal samples exposed to DI water at 15 MPa pore pressure. The observed behavior may be controlled by competition between dilatant granular flow and compaction enhanced by the presence of water. Together with our previous work on frictional properties of coal-shale mixtures, our results imply that the presence of a weak, coal-dominated patch on faults that cut or smear-out coal seams may promote unstable, seismogenic slip behaviour, though the importance of this in enhancing either induced or natural seismicity depends on local conditions.

**Keywords**: slip-weakening, rate-dependent friction, frictional healing, strain localization, coal maturity, coal molecular structure

# 1 Introduction

Carbonaceous materials (e.g. amorphous carbon, graphite, organic matters) are widely present in the lithosphere, including in several large fault zones over the world (Kaneki and Hirono, 2019), such as Longmenshan thrust belt in China (Kuo et al., 2014), Atotsugawa fault zone in Japan (Oohashi et al., 2012) and Alpine fault zone (Kirilova et al., 2017). As is well known that graphite has very low frictional strength and that amorphous carbon or organic matters can be transformed into graphite at a seismic slip due to so-called graphitization process. The presence of carbonaceous materials may, therefore, act as a lubricant to play a key role in frictional properties and accordingly in promoting instability of the fault (Oohashi et al., 2011, 2013; Kuo et al., 2014). In the meanwhile, organic-rich rocks (such as coal, shale and clay), as main source rocks for (un)conventional natural gas, may also play a role in induced seismicity upon gas production (e.g. Kohli and Zoback, 2013; Liu et al., 2020), water injection (e.g. Ellsworth, 2013) and coal mining (e.g. Westbrook et al., 1980). Compared to graphite (Ruan and Bhushan, 1994; Moore and Lockner, 2004; Kirilova et al., 2018), however, limited experimental data on frictional properties of organic-rich rocks (Liu et al., 2020), particularly on coal, under in situ PT conditions is reported. Although coal has been widely investigated because of its importance in fuel energy and industry (Guo et al., 2018; Chen et al., 2019), its frictional properties are not yet well determined and understood. This contribution address frictional properties of coal and the likely mechanisms.

We first focus on coal structure and graphitization process seen in experiments. Many techniques, such as transmission electron microscope (TEM), nuclear magnetic resonance (NMR), X-ray diffraction (XRD), Raman spectroscopy, and Fourier Transform Infrared spectroscopy (FTIR), have been applied to determine coal structure, because of its complexity and heterogeneity (Li et al., 2015a, 2015c; Baysal et al., 2016; Song et al., 2019). From a chemical point of view, coal, in general, mainly consists of aromatic layers where aromatic nucleus are surrounded by peripheral aliphatic chains and oxygen functional groups (Mathews and Chaffee, 2012; Ahamed et al., 2019). Lu et al. (2001), based on XRD analysis of Australian coals ranging in rank from high volatile bituminous to semi-anthracite, proposed a simplified model for describing coal molecular structure. This model suggests coal consists of both amorphous (non-aromatic structures) and crystalline (a condensed, layered aromatic structure) forms of carbon. The aromatic layers in coal may become more ordered or more uniform packing during deformation to form graphite under experimental conditions of a constant high confining pressure of 500 MPa, deviatoric stresses, variable temperatures of 300–600 °C and strains to 33% (Ross and Bustin, 1990; Ross et al., 1991). Ross and the co-authors (1990; 1991), based on their experiments, reported that the shear strains associated with strain energy can drastically lower the activation energy and accordingly facilitate the graphitization process. Similarly, molecular dynamics simulations of sliding at the interface between amorphous carbon and diamond films, at a rate of 10 m s$^{-1}$, performed by Ma et al. (2014), show that covalent bond reorientation, phase transformation and structural ordering preferentially occur in localized bands in amorphous carbon film, and that this shear localization causes weakening. Apart from high pressure and temperature experiments, the graphitization process has also seen in high-velocity friction

experiments. Oohashi et al. (2011), for example, performed friction experiments on both amorphous carbon and graphite using a rotary shear apparatus under conditions of normal stress of 0.5–2.8 MPa and slip rates of 50 μm s$^{-1}$ – 1.3 m s$^{-1}$ in atmospheres of air and nitrogen. Their experiments showed a) a steady-state friction coefficient of 0.54 for amorphous carbon at slow slip rates versus 0.1 for graphite at all slip rates, b) major slip-weakening of the amorphous phase, at slip rates >10 mm s$^{-1}$, to a steady-state μ-value of 0.1, and c) XRD and TEM evidence of graphitization of the amorphous carbon during shear at high slip rates. The authors suggested that large shear strains, short-lived flash heating, and/or stress concentrations at asperity contact points may cause graphitization of amorphous carbon, even at low temperatures and pressures under anoxic environments. Similar friction experiments, performed by Kuo et al. (2014) on natural samples collected from the 2018 Wenchuan earthquake slip zone, also showed graphitization of carbonaceous minerals due to frictional heating at seismic slip rates. On the other hand, Kirilova et al. (2018) performed double direct shear experiments on dry synthetic graphitic carbon at slow slip rates of 1-100 μm s$^{-1}$ and normal stresses of 5 and 25 MPa, at room temperature. They found slip weakening of the samples from a peak frictional strength of ~0.4-0.55 to a steady-state value of ~0.15-0.25, which is higher than the steady-state μ-value seen in high-velocity friction experiments on graphite. Their TEM and Raman observations suggest shear-enhanced structural disorder with increasing shear strain developing in localized slip zones. In addition, Ruan and Bhushan (1994) investigated frictional properties of highly oriented pyrolytic graphite using a friction force microscope and TEM, and found that the friction coefficient of the well-ordered carbon of (0001) plane is much smaller compared with that of the randomly-ordered carbon. These indicate internal carbon crystal structural difference may lead to a significant difference in frictional strength of graphite materials.

We now return to frictional properties of coal. O'Hara et al. (2006) performed high-velocity (1 m s$^{-1}$) friction experiments on high volatile bituminous coal at a normal stress of ~0.6 MPa, employing a large displacement (maximum of ~80 m). Their results demonstrated significant slip weakening behaviour and enhanced coal maturity. Specifically, the friction coefficient decreased from 0.8-1.2 to 0.1-0.4 and random vitrinite reflectance increased from ~0.6% to ~0.8%. Besides the thermal effect of shear heating, they suggested coal gasification, fluctuations in fluid pressure and gas pressurization also played a role in determining the frictional behaviour. Similar coal maturity evolution caused by frictional heating was also reported by Kitamura et al. (2012). More recent research reported by Kaneki and Hirono (2019) investigated the frictional strength of lignite, bituminous coal, anthracite and graphite by performing high-velocity (1 m s$^{-1}$) rotary-shear friction experiments at room temperature. They found that the peak frictional strength, for all samples, decreased with increasing maturity from 0.5 to 0.2, and the marked dynamic weakening was observed for lignite, bituminous coal and anthracite, from peak friction coefficient values of ~0.3–0.5 to dynamic values of ~0.1–0.2. TEM, IR and Raman observations performed on the samples before and after frictional shearing suggested that the marked dynamic weakening behaviour observed in lignite, bituminous coal and anthracite was caused by a shear-induced graphitization process possibly dominated by flash heating (Kaneki and Hirono, 2019). Somewhat different results were obtained by Fan and Liu (2019). These authors performed low-velocity direct shear / friction experiments on pre-cut coal samples (low volatile bituminous coal) exposed to various fluids (helium,

carbon dioxide, water, and moisturized methane) at a constant effective normal stress of 2 MPa, employing shear rates of 1–10 µm s⁻¹. Their results showed a) no slip-weakening, b) a steady-state friction coefficient for samples exposed to water and moisturized methane of ~0.15, c) a much higher friction coefficient in samples exposed to helium (~0.53) and carbon dioxide (~0.43), and velocity strengthening behaviour, regardless of the fluids. By contrast, we performed low-velocity (i.e. 0.1–100 µm s⁻¹) direct shear experiments to investigate frictional properties of simulated fault gouges prepared from coal-shale mixtures under (near) in situ PT conditions (i.e. effective normal stress of 40 MPa and 100 ℃) (Liu et al., 2020). We found that only the samples with coal volume fraction ≥50%, including pure coal, showed marked slip-weakening behaviour, from the peak value of ~0.47 to (near) steady-state value of ~0.30, regardless of the employed experimental conditions. Interestingly, such slip-weakening is limited to small initial displacements (2–3 mm), and does not occur during slip reactivation. We, based on the limited microstructure observation, inferred this slip-weakening was caused by strain localization in coal-rich shear bands, accompanied by a change in coal molecular structure, as opposed to the graphitization effects seen in high-velocity friction experiments. As the main aim of our recent research (Liu et al., 2020) was to investigate the effects of coal content on frictional properties of Carboniferous shale in the context of induced seismicity in Carboniferous source rocks below Europe's largest gas field, we only reported one velocity stepping friction experiments on pure coal. As a result, more experimental research is needed to better understand frictional properties (such as frictional strength, rate-dependent friction and frictional healing) of coal, sheared at slow slip rates under in situ PT conditions, accompanied by the development of coal molecular structure upon shear deformation.

In this paper, we investigate the frictional behaviour of deep natural coal (as a source rock for (un)conventional natural gas) under near in situ conditions. This was achieved by performing the friction experiments on simulated fault gouges prepared from bituminous coal collected from the upper Silesian Basin of Poland. We performed velocity stepping and slid-hold-slide experiments under both vacuum dry and wet conditions, at a constant temperature of 100 °C, employing sliding velocities of 0.1–100 µm s⁻¹ and effective normal stresses ranging from 25 to 45 MPa. Data on the frictional strength and rate dependence of friction are documented and a full Rate and State Friction (RSF) description is derived. In an attempt to understand the likely mechanisms determining frictional behaviour, post-test analysis was performed on both unsheared and sheared samples using microstructural observation, micro-area X-ray diffraction and Raman spectroscopy. Data on crystal structure parameters and Raman parameters are also obtained. Finally, we discuss the implications of our findings for understanding frictional strength and seismic potential of coal-rich faults.

## 2 Experimental methods

### 2.1 Approach

Following Hunfeld et al. (2017) and Liu et al. (2020), we performed direct shear experiments to measure the frictional sliding strength and rate-dependent friction of simulated coal fault gouges at near in situ PT conditions for deep coal seams.

And then we apply the rate and state friction (RSF) approach to determine the rate dependence of friction. Post-test analysis by means of microstructural observation, micro-area XRD and Raman spectroscopy were performed on the deformed gouge samples, in an attempt to understand the observed frictional behaviour.

## 2.2 Sample materials

The coal samples used in this study were prepared from natural high volatile bituminous coal with total organic carbon (TOC)
of 69.6 wt% , collected from Brzeszcze Mine (Seam 364) in the upper Silesian Basin of Poland (Hol et al., 2011; Liu et al., 2020). Petrological and chemical analyses reported by Hol et al. (2011) showed that the bituminous coal has a vitrinite reflectance of $0.77 \pm 0.05$ % and has a vitrinite content of 60.1 wt%, alongside liptinite 9.8 wt%, and inertinite 30.1 wt%. Furthermore, it contains 74.1 wt% carbon, 5.3 wt% hydrogen, 1.4 wt% nitrogen, 0.7 wt% Sulphur, 18.5 wt% oxygen, as well as 2.9 wt% moisture and 5.2 wt% ash (mineral) content. Raw coal sample was crushed to obtain powder with a grain size of
<50 µm. For each experiment, a gouge layer with a thickness of ~1 mm (see details in Table 1) was prepared by compacting coal powders in a purpose-made die, at ~20 MPa for ~2 minutes. The gouge layer was then assembled into a "direct shear" assembly, comprising two opposing L-shape pistons, designed for direct shear testing in a triaxial deformation apparatus (following Samuelson and Spiers, 2012). Note that we marked starting, loose coal powders and one coal gouge sample without shear deformation as S* and S0, respectively (see Table 1). They are used as the control samples for XRD and
Raman tests, in an attempt to determine the effects of the shear/friction processes on the molecular structure of carbon in coal. Note that the gouge sample S0 was prepared by compacting at ~20 MPa for ~15 h at 100 ℃.

## 2.3 Direct shear experiments and post-test sample treatment

We performed 8 direct shear experiments at a constant temperature of 100 ℃ using a conventional triaxial testing machine (referred to as the Shuttle Machine, see Verberne et al., 2014a), equipped with the direct shear assembly described above. An
150 independent ISCO 65 volumetric (syringe) pump was used to control pore fluid pressure. A detailed description of the machine was given by Verberne et al. (2014a) and Hunfeld et al. (2017). The experiments employed confining pressures ($\sigma_n$) of 40, 50, 55 and 60 MPa, and a constant pore fluid pressure of $P_f$=15 MPa or under vacuum (i.e. $P_f$=0). Distilled (DI) water was used as pore fluid for experiments S3–S8, while the experiments S1 and S2 were tested under vacuum dry (see Table 1 for details of experimental conditions). In each experiment, the sample assembly, initially drained to the lab air, was first
heated to ~100 ℃ at a confining pressure of ~20 MPa, and left to equilibrate for ~15 h (overnight). Then the pore fluid was introduced into the sample and pressurized to 15 MPa at a confining pressure of ~20 MPa. The confining pressure was subsequently increased to a certain value and left the system for ~3 h to equilibrate before shearing. Seven velocity-stepping (VS) and one slide-hold-slide (SHS) experiments were conducted in this study (see Table 1). In the VS experiments, samples were sheared at a constant velocity (V) of 1 µm s$^{-1}$ for ~2.5 mm shear displacement, after which the loading rate was
instantaneously stepped in the range 0.1–100 µm s$^{-1}$ over total displacement up to almost 6 mm. SHS experiment was also

performed at a constant velocity (V) of 1 μm s$^{-1}$ interrupted by hold intervals in the range 300 to 30000 s, in an attempt to determine the healing effects of coal.

After each experiment, the direct shear setup was dismantled, and intact fragments of the sheared gouge layers were recovered and oven-dried for several days. Note that, for the observation using scanning electron microscope (SEM), to avoid the potential problem caused by use of carbon-bearing epoxy, no special treatment was performed on the samples. Microstructure of the sheared samples S1–S8 were observed using an optical microscope and SEM. For each sample, we carefully chose the fragments that have clear, clean slip surface, and the fragments that have relatively flat cross-section in an orientation parallel to the shear direction and perpendicular to the shear plane (e.g. Fig. 1a). Note that artificial microfractures formed during extraction of the samples from the experimental apparatus and subsequent treatment can be easily recognized and excluded. Micro-area XRD and Raman spectroscopy analysis were performed on principal slip zone (PSZ) and weakly deformed zone (WDZ, terminology following Oohashi et al., 2011) of the samples S1–S8 (e.g. Fig. 1c). Note that the surface of WDZ in Fig. 1c was exposed by scraping PSZ using abrasive paper. Recall that the samples S* and S0 (i.e. without shear deformation), as control experiments, were also tested using XRD and Raman spectroscopy. The location of the micro-area was selected randomly in PSZ, WDZ or unsheared surface, and was schematically marked in Fig. 1b and Fig. 1c (blue circles).

### 2.4 Post-test analysis

### 2.4.1 Microstructural methods

A Leica EZ4w optical stereomicroscope and a table-top SEM fitted with an Energy Disperse Spectroscopy (EDS) were used to investigate the microstructure of the fragments retrieved from the deformed samples. Note that the fragments were not coated because coal samples have sufficient electroconductivity. The samples were imaged in the secondary electron mode, using an acceleration voltage of 15–20 kV. Besides, EDS was used to determine whether the observed grains are coal component or other minerals.

### 2.4.2 Micro-area XRD

We performed the micro-area XRD experiments on samples S0–S8 (except S2 and S6), in an attempt to determine the crystal structure of coal samples. Sample S* was tested by X-ray powder diffraction in a conventional mode. These were achieved using the Smartlab 9 kW X-ray diffractometer with a Cu target at ambient temperature. Samples were scanned in 2θ range from 10 to 65° at a rate of 1° min$^{-1}$. The micro-area (~300 μm in diameter) in the PSZ and WDZ was measured for each sheared sample (see Fig. 1c). Note that the sample S*, S0 and S5 were measured twice for examining data reproducibility. For those samples, we took the average values as the parameters and the standard deviation as error bars.

### 2.4.3 Raman spectroscopy

We performed Raman measurements on samples S*–S8 to determine the development of coal maturity upon the shear/friction experiments. This was done using a Renishaw inVia™ laser Raman instrument (with a spectra resolution of 1 cm$^{-1}$) that was connected to a Leica DMLM microscope. The 514.5 nm argon-ion green laser was used for all experiments. The laser was focused through a ×50 objective, with a laser spot size of ~c.2 μm. We used a laser power of 0.2–1.0 mW (1% – 5% of ~17 mW full power) to avoid thermal damage on the targeted surface of coal samples. The scan range was limited to 50–3000 cm$^{-1}$, in order to assess the first-order region (900–2000 cm$^{-1}$) and part of the second-order region (2200-3300 cm$^{-1}$). For each scan, we set the acquisition time of 10 s and the cumulative times of 3–5. For the unsheared and sheared samples (S0–S8), we measured three points randomly distributed in the unsheared or principal slip surface for each sample. Note that for the sample S4 only, we also measured 3 points in the weakly deformed zone. We accordingly took the mean values as the representative Raman parameters and the standard deviations as error bars for each sample. Note that we measured only one point in the powdered sample S*.

### 2.5 Data acquisition, processing and analysis

### 2.5.1 Mechanical data acquisition and treatment

Internal axial force, confining pressure, pore fluid pressure, sample temperature and loading piston displacement were measured in each experiment and the signals logged using a 16-bit National Instruments AD converter and logging system (for details, see Hunfeld et al., 2017). Following Liu et al. (2020) and Hunfeld et al. (2017), the data were processed to yield sample shear stress versus shear displacement data corrected for machine stiffness (see details in Liu and Hunfeld, 2020). The frictional strength of the samples was characterised by defining the apparent coefficient of sliding friction ($\mu$) as the ratio of sample shear stress ($\tau$) over the effective normal stress ($\sigma_n^{eff}$), assuming zero cohesion.

$$\mu = \frac{\tau}{\sigma_n^{eff}} \tag{1}$$

where $\sigma_n^{eff} = \sigma_n - P_f$. Here $\sigma_n$ represents the normal stress or confining pressure employed in the experiments, and $P_f$ represents the pore fluid pressure.

The rate dependence of friction was quantified using the RSF theory (Dieterich, 1979; Ruina, 1983), coupled with the empirical Dieterich-type "aging law" (e.g. Marone, 1998):

$$\mu = \mu_0 + a\ln\left(\frac{V}{V_0}\right) + b\ln\left(\frac{V_0\theta}{D_c}\right) \tag{2}$$

$$\frac{d\theta}{dt} = 1 - \frac{V\theta}{D_c} \tag{3}$$

which describes the evolution of friction coefficient $\mu$ from a reference steady-state value ($\mu_0$) towards a new steady-state value, over a critical slip distance ($D_c$), in response to an instantaneous change in sliding velocity from an initial sliding velocity ($V_0$) to a new sliding velocity ($V$). The state variable $\theta$, which describes the evolution of gouge friction via Eq. (3), is

commonly viewed as the average lifespan of a population of grain-to-grain contacts (Marone, 1998). At steady state, i.e. when $d\theta/dt = 0$, Eq. (2) reduces to:

$$(a - b) = \frac{\mu - \mu_0}{\ln(V/V_0)} \qquad (4)$$

where the parameter (a-b) reflects the rate-sensitivity of friction coefficient. From an RSF point of view, if fault rocks exhibit an increase in frictional strength upon increased sliding rate, i.e. velocity strengthening behaviour with (a-b)>0, they are not prone to generating accelerating slip and are termed conditionally stable (Scholz, 2019). On the other hand, when the frictional strength of a fault rock decreases upon increased sliding rate, the fault rock exhibits velocity weakening behaviour with (a-b)>0. Given sufficient elastic compliance in the loading system, this behaviour can cause repetitive slip instabilities, or stick-slip events, viewed as the laboratory equivalent of earthquakes (Brace and Byerlee, 1966; Marone, 1998; Scholz, 1998). Here, we solve Eq. (2) accompanied with Eq. (3) simultaneously with an equation describing the elastic interaction with the testing machine via the stiffness, using Eq. (1) as a constraint. The values for $a$, $b$ and $D_c$ can then be obtained as the solutions of a nonlinear inverse problem using an iterative least-squares minimization method (Ikari et al., 2009), thereby obtaining a full RSF description of the material from our experiments. In performing RSF inversion, departures from steady-state frictional sliding were corrected using linear detrending of hardening or softening behaviour (see Fig. 4a), and thus the slope of linear detrending ($\eta$) was obtained. A detailed description was also given by Blanpied et al. (1998) and Ikari et al. (2013).

**2.5.2 Determining crystal structure parameters from XRD data**

We first corrected for the background noise of the resulting diffractograms using a spline curve (see Fig. 2), obtaining an approximative profile of crystalline carbon (i.e. the background-subtracted intensity profile shown in Fig. 2). The obtained profile was further deconvoluted using Lorentzian and Gaussian function to determine the crystal structure parameters of coal. Specifically, Lorentzian peaks were first employed to determine minerals in $2\theta$ range of ~16–30° (10–30° for powdered sample S*) and ~35°–58°, in an attempt to remove the mineral peaks from the background-subtracted curve. We then employed three Gaussian peaks to fit the background-subtracted, mineral-free profile at around 20°, 26° and 43°, obtaining $\gamma$-band, 002-band and 10-band, respectively (see Fig. 2). In general, the $\gamma$-band reflects the structure of ring-free saturated hydrocarbons (see detailed description in Yen et al., 1961), whereas the (002) and (10) bands reflect the ring structure of the aromatic layers of crystalline carbon (Lu et al., 2001). The fitting parameters, such as peak position ($\theta$), full width at half maximum ($\beta$) and area ($A$), were obtained. The structure parameters of carbon crystallite in coal, such as interlayer spacing ($d_{002}$), crystallite stacking height ($L_c$), crystallite diameter ($L_a$), were determined using the empirical Braggs and Scherrer equations (Eqs. 5–7). Note that we took the mean values as the parameter for samples S*, S0 and S5.

$$d_{002} = \frac{\lambda}{2\sin\theta_{002}} \qquad (5)$$

$$L_c = \frac{0.89\lambda}{\beta_{002}\cos\theta_{002}} \qquad (6)$$

$$L_a = \frac{1.84\lambda}{\beta_{10}\cos\theta_{10}} \tag{7}$$

where $\lambda$ is the wavelength of the applied X-ray (0.154056 nm for Cu Kα radiation); $\theta_{002}$, $\beta_{002}$, $\theta_{10}$ and $\beta_{10}$ represent the peak position of and full width at half maximum of bands (002) and (10) respectively. Theoretically, the areas of the bands (002) and (γ) ($A_{002}$ and $A_\gamma$) are believed to be equal to the number of aromatic and aliphatic carbon atoms respectively (Yen et al., 1961), so that the aromaticity ($f_a$) of the samples can be estimated using Eq. (8). Coal rank can also be assessed using the ratio of maximum intensity of (002) over that of (γ) band ($I_{002}/I_\gamma$, as seen in Eq. (9)).

$$f_a = \frac{A_{002}}{A_{002}+A_\gamma} \tag{8}$$

$$\text{Coal rank} = \frac{I_{002}}{I_\gamma} \tag{9}$$

### 2.5.3 Determining Raman parameters

Raman spectroscopy is a powerful tool for analyzing the information of the molecular structure of organic matters (Ulyanova et al., 2014). It is well known that G band, which is located around 1580 cm$^{-1}$ in Raman spectra, is the only peak in the first-order region (i.e. 900–2000 cm$^{-1}$) for pure single graphite crystal. D band is another peak located around 1350 cm$^{-1}$ in Raman spectra, which is generally present along with G band for other carbon materials such as graphite with defective lattice, activated carbon and coal (Tuinstra and Koenig, 1970; Potgieter-Vermaak et al., 2011; Childres et al., 2013). The Raman parameters for coal include the peak position, full width at half maximum (FWHM) and intensity of G and D band, the Raman band separation (RBS = G-position - D-position), the intensity ratio of D-band over G-band ($I_D/I_G$) and the saddle index (SI = the intensity of G-band divided by that of the saddle). These parameters can be obtained by several processing methods (Beyssac et al., 2003; Sadezky et al., 2005; Wilkins et al., 2014; Henry et al., 2018; Khatibi et al., 2018). In this paper, we use the method proposed by Henry et al. (2018) to determine the Raman parameters, as it has been well tested for Carboniferous organic-rich mudstones and coals (Henry et al., 2019). In general, Henry's method includes: a) The raw Raman spectra were first smoothed using a Savitzky-Golay filter, i.e. a 21-point quadratic polynomial algorithm; b) A 3rd-order polynomial function was then used to correct for baseline; c) The smooth, baseline-removed spectra were finally normalized to a common G-band height of 2000 arb. units. Specifically, we used the automated Microsoft Excel® spreadsheet proposed by Henry et al. (2018) to process our spectra data obtained at 900–2000 cm$^{-1}$.

## 3 Results

### 3.1 Mechanical data

### 3.1.1 Frictional strength of simulated coal fault gouges

Typical apparent friction coefficient ($\mu$) versus displacement data obtained in velocity stepping experiments (Exp. S1–S7) are plotted in Fig. 3. All experiments plotted in Fig. 3a showed rapid, near-linear initial loading up to a peak friction

coefficient at a shear displacement of ~0.6 mm, followed by sharp, post-peak slip weakening from peak values of ~0.48 to a near (quasi) steady-state value of ~0.3 at a shear displacement of ~2.2 mm. The quasi steady-state friction coefficient decreased slightly with displacement reaching a new quasi-steady state value at 4-6 mm. This slight weakening might be caused by the reduction of the load supporting area of the sample during shear deformation. Nevertheless, to quantify this

effect, we define $\mu_{peak}$ as the peak friction coefficient obtained at 0.5–0.75 mm shear displacement, and we take $\mu_{ss1}$ and $\mu_{ss2}$ to represent the near steady-state friction coefficient values obtained at ~2.2 mm and ~5.7 mm shear displacement respectively. These friction coefficient data ($\mu_{peak}$, $\mu_{ss1}$ and $\mu_{ss2}$) are plotted in Fig. 3b as a function of effective normal stress, which indicates that they are more or less independent of effective normal stress. The largest values for $\mu_{peak}$ (0.524) and $\mu_{ss1}$ (0.338) were obtained in experiment S3, which was performed at a confining pressure of 40 MPa and a pore water pressure

of 15 MPa (i.e. at effective normal stress of 25 MPa). The values of $\mu_{peak}$ and $\mu_{ss1}$ obtained at vacuum dry conditions (i.e. $\mu_{peak}$ of ~0.48 and $\mu_{ss1}$ of ~0.30 for experiments S1 and S2) are slightly higher than those ($\mu_{peak}$ = ~0.46 and $\mu_{ss1}$ = ~0.28) obtained for the samples S5 and S6 that were exposed to DI water at the same effective normal stress of 40 MPa. All frictional strength data are summarized in Table 1, including $\mu_{peak}$, $\mu_{ss1}$ and $\mu_{ss2}$.

**3.1.2 Rate dependence of friction**

The individual RSF parameters $a$, $b$ and $D_c$, and the rate-sensitivity parameter ($a$-$b$), obtained in all velocity-stepping experiments, using a full RSF inversion approach, are summarized in Table 2. The slope $\eta$ of the linear slip weakening trend obtained after each upward velocity step in the displacement interval of 2-4 mm in experiments S1-S7 is also listed in Table 2. The ($a$-$b$) data obtained for upward steps are plotted in Fig. 4b as a function of effective normal stress. Here we plot upward stepping data only, as it is these that are most relevant to rupture nucleation (Marone, 1998). Almost all ($a$-$b$) values

fall in the range of -0.006 to +0.002, and systematically decrease in the higher velocity steps in all samples (see Fig. 4b). It is also clear from Fig. 4b that ($a$-$b$) values are insensitive to effective normal stress, but sensitive to velocity and pore fluid condition. Specifically, all samples show velocity strengthening at velocities stepped from 0.1 to 1 $\mu$m s$^{-1}$ (where $a$-$b$>0), but velocity weakening in steps from 10 to 100 $\mu$m s$^{-1}$ (where $a$-$b$<0). For velocity steps from 1 to 10 $\mu$m s$^{-1}$, sample S2 tested under vacuum dry conditions shows velocity weakening (i.e. $a$-$b$<0), while samples tested with DI pore water at a pressure

of 15 MPa show velocity strengthening, except for sample S5 which exhibits velocity-weakening.

In addition to the above treatment of the RSF data, the slope $\eta$ of the linear slip weakening portions of the friction vs. displacement curves was plotted in Fig. 4c as a function of the up-step velocity. This shows that the absolute magnitude of $\eta$ systematically increases with increasing slip rate, reflecting velocity-enhanced slip-weakening behaviour. We note that this

type of linear slip weakening behaviour has also been observed in velocity-stepping (0.03–100 $\mu$m s$^{-1}$) experiments performed on natural fault gouge (37–65% clay minerals, up to 40% quartz + plagioclase and little calcite) collected from the Nankai subduction zone in Japan, and has been put forward as a mechanism for promoting slow earthquakes (see details in Ikari et al., 2013). This behaviour seen in Fig. 4c, may warrant deeper investigation in future.

### 3.1.3 Frictional healing effects

The slide-hold-slide loading path data (Exp. S8) shown in Fig. 5a indicates a clear but minor strength recovery or healing effect ($\Delta\mu$) upon re-shear, followed by slip-weakening to achieve a new quasi-steady state. The magnitude of restrengthening ($\Delta\mu$) increases with the logarithm of hold time ($t$), and is well described by the equation $\Delta\mu = \beta\log(1+t/t_c)$ (e.g. Marone, 1998), where $\beta = 0.006 \pm 0.001$ and $t_c = 9 \pm 9$ s (Fig. 5b).

### 3.2 Microstructure of the deformed coal gouge

The representative microstructure for a sliding surface of sample S6 obtained using an optical microscope in a reflected light mode is shown in Fig. 6a, indicating a highly reflective (mirrorlike) area located in the left half of Fig. 6a. This may be similar to the reported mirror slip surface (Siman-Tov et al., 2013; Fondriest et al., 2013; Verberne et al., 2014b). Besides, a principal boundary slip zone (PSZ, of ~15–25 µm thick) accompanied with weakly deformed zone (WDZ) was observed in all deformed coal gouges (see Fig. 6b for a representative reflected light micrograph of the sample S1 in an orientation

parallel to shear direction). Note that unlike boundary shear bands that were observed in all samples, R- and Y- shear bands were only observed in the sample S5 (see Fig. 6c and 6d). This may be because the surface of the fragments chosen from other samples were not flat enough to capture R- and Y- shear bands. Besides, SEM secondary electron images shown in Fig. 7a–e, may indicate the development of first microfractures inside the starting coal grains (~50 µm), then failure to form small grains (<10 µm) and finally shear bands. This likely reflects the process of cataclasis or granular flow during shear

deformation in coal gouge (Niemeijer and Spiers, 2007; Verberne et al., 2014b; Hadizadeh et al., 2015). More importantly, a marked stack-layer structure was clearly observed at the margin of PSZ (see Fig. 7f), likely reflecting an interaction between PSZ and WDZ and the role of PSZ during the friction process. EDS data measured at the three representative spots located in Sample S4 (see Fig. 7b,c and f) are shown in Fig. 7g, indicate high C and O but little mineral content in the WDZ and PSZ zone.

### 3.3 Development of coal crystal structure (XRD data)

    The raw X-ray diffractograms of the coal samples S*–S8 are presented in Fig. 8. Note that the diffractogram for coal powdered sample (S*) was scaled to a comparable size with other samples. Note also that the minor peaks observed in Fig. 8 represent minerals (such as kaolinite and dolomite) in coal samples. Fig. 8 shows that the (002), (10) and γ-side bands general characteristic of coal were observed in all samples (following Hirsch, 1954 and Lu et al., 2001). In addition, all

samples showed a high background intensity, indicating that a significant proportion of amorphous carbon (i.e. non-aromatic component) was present in our coal samples (Dun et al., 2013). This high background is characteristic of materials having non-uniformly developed crystal structures (e.g. coal), regardless of the specimen holder or diffractometer used in the experiments (e.g. Li et al., 2015a; Baysal et al., 2016). Importantly, Fig. 8 demonstrates an apparent difference between unsheared (S* and S0) and sheared (S1–S8) samples, compared to that minor or little difference was observed between the

sheared samples. This strongly suggests the effects of the shear/friction on the development of molecular structure in coal. We also note that no graphite was formed after the shear/friction processes, despite the PSZ and/or WDZ of samples S3, S5 and S8 have a strong peak (26.29°) that is very close to the graphite (002) peak (26.38°).

Note that in this study we assume the crystalline carbon in coal, in general, consists of graphite-like layered aromatic structures plus marginal aliphatic structure (Lu et al., 2001). All structure parameters for carbon crystallite obtained using the methods described in Sect.2.5.2 are listed in Table 3, and the representative structure parameters obtained from the samples S*–S8 were also plotted as a function of apparent steady-state shear stress (Fig. 9a) and effective normal stress (Fig. 9b) measured at the shear displacement of ~5.7 mm in the direct shear experiments. Specifically, the interlayer spacing ($d_{002}$) of the layered graphite-like structure in the sheared coal gouges yields 3.47–3.53 Å, including PSZ and WDZ, which is lower than ($d_{002}$) of 3.56–3.58 Å obtained for the unsheared samples S* and S0, as seen in Table 3 and Fig. 9a. This suggests the layered graphite-like structures in coal, i.e. condensed aromatic system, became more condensed after the shear/friction experiments. Careful inspection of Fig. 9a also indicates a thinner interlayer spacing ($d_{002}$) measured in PSZ compared to that measured in WDZ for most samples, suggesting the development of the layered graphite-like structures in boundary shear band. The stacked height of crystalline carbon (i.e. $L_c$) in coal increased from ~13 Å to ~20 Å, while the diameter of crystalline carbon (i.e. $L_a$) decreased from 20–21 Å to 15–19 Å. We also note that, for most samples, the values of $L_c$ and $L_a$ measured in WDZ lie in between that measured in the unsheared samples and PSZ of the sheared samples. Meanwhile, aromaticity ($f_a$), i.e. the ratio/fraction of aromatic carbon atoms, yields 0.44–0.49, 0.43–0.64 and 0.53–0.67 for the unsheared samples, WDZ and PSZ of the sheared samples, respectively. The parameter ($I_{26}/I_{20}$), representative of coal rank, accordingly increased from 1.21–1.29 to 1.33–2.29 after the shear/friction experiments. These clearly indicate the development of the condensed aromatic system in coal upon the shear/friction processes. Besides, it is seen from Fig. 9 that the structure parameters measured in the sheared samples, including PSZ and WDZ, seem to be insensitive to the applied effective normal stress and the apparent steady-state shear stress measured at the shear displacement of ~5.7 mm.

### 3.4 Results of Raman analysis

The averaged normalized spectra of all samples after smoothing and background correction is plotted in Fig. 10. The D and G bands were observed at ~1360 $cm^{-1}$ and ~1600 $cm^{-1}$ for all samples, respectively. Both D and G band observed in the sheared samples are more narrow than those observed in the unsheared samples. Also, the D band observed in the sheared samples slightly shifted to the left, while the G band slightly shifted to the right, compared to those observed in the unsheared samples. This likely reflects an increase in maturity of coal after the shear/friction experiments. The Raman parameters described in Sect.2.5.3 were obtained from Fig. 10 and listed in Table 4. As shown in Fig. 11a and 11b, the Raman parameters for the samples S0 and PSZ of S1–S8 were also plotted as a function of apparent steady-state shear stress and effective normal stress measured at the shear displacement of ~5.7 mm in the direct shear experiments. It shows, in general, an obvious difference in the Raman parameters between the sheared coal samples (S1–S8) and the unsheared sample

S0, suggesting the role of the shear/friction. Particularly, G-FWHM values decrease from 91.3 cm$^{-1}$ measured in the unsheared sample to 71.7–79.7 cm$^{-1}$ measured in the sheared samples. RBS and SI values increase from 227.7 cm$^{-1}$ and 3.27 measured in the unsheared samples to 239.3–256.7 cm$^{-1}$ and 3.65–4.26 measured in the sheared samples. By contrast, we found similar parameter values of $I_D/I_G$ for all samples, yielding 0.543–0.561, which is not sensitive to the shear deformation. It is also seen from Fig. 11a and 11b that all Raman parameters measured in the sheared samples are not sensitive to the applied effective normal stress and apparent steady-state shear stress. Recall that we measured Raman spectra in WDZ only for the sample S4. We plotted the Raman parameters measured in samples S*, S0 and S4 in Fig. 11c, indicating a slight difference between WDZ and PSZ, as well as the difference between S* and S0.

## 4 Discussion

All experiments show significant slip-weakening behaviour, from a peak friction coefficient of ~0.5 to a near steady-state value of ~0.3. Post microstructure observation indicates the development of shear bands. Besides, XRD and Raman analysis suggest the shear deformation may change the molecular structure and maturity of coal in the shear bands. Furthermore, the VS experiments showed that a) little effect of effective normal stress on frictional strength and (*a-b*) values were found; b) (*a-b*) values systemically became smaller at higher velocity steps; c) the samples exposed to DI water at a pore pressure of 15 MPa exhibited velocity strengthening behaviour at velocity steps of 0.1–10 μm s$^{-1}$, but velocity weakening behaviour at velocity steps of 10–100 μm s$^{-1}$, as opposed to the sample under vacuum dry that showed velocity weakening behaviour at almost all velocity steps employed in this study. The SHS experiment demonstrated minor frictional healing (β=0.006 ± 0.001) in a water-saturated coal gouge sample. In the following, we first attempt to elucidate the development of the molecular structure of coal upon shear deformation. We then discuss whether the shear-induced molecular structural change dominated the marked slip-weakening behaviour observed in coal gouges. We also discuss the velocity or rate-dependent friction of coal. Finally, we consider, in a broad way, the implications of our findings for the frictional strength and (induced) seismic potential of coal-rich or organic-rich faults.

### 4.1 Development of molecular structure and maturity of coal in shear bands

Following Lu et al. (2001), we assume our bituminous coal consists of graphite-like crystalline (i.e. a condensed aromatic system) and amorphous (i.e. non-aromatic system) forms of carbon. Our XRD results on the unsheared samples (S* and S0) show the structure of the graphite-like crystalline carbon in bituminous coal, yielding the interlayer spacing ($d_{002}$) of ~3.56 Å, the crystallite diameter ($L_c$) of ~13.5 Å, the stacked height ($L_a$) of ~19 Å, and aromaticity ($f_a$) of ~0.47. These parameter values are consistent with those of the similar bituminous coal reported by Li et al. (2015a), Okolo et al. (2015) and Zhang et al. (2015). Note that the parameter values of $L_c$ and $L_a$ that were calculated using Scherrer's equation may be larger than the real size (Lu et al. 2001) and that the presence of mineral peaks particularly in the 2θ range of 27–50° may also influence the fitting process (i.e. fitting Gaussian peak to (002), (γ) and (10) band), and may accordingly influence the accuracy of the

structure parameter values. However, these influences should be consistent for all samples. We, therefore, believe the values of the structure parameters shown in Table 3 and Fig. 9 could be influenced by these factors as a systemic error and accordingly would not change the trend or results that we observed in Fig. 9. Furthermore, the minor standard deviations shown in the measurements on samples S*, S0 and S5 indicate a good reproduction of our XRD measurements. As a result, we believe that our XRD results on both unsheared and sheared samples, particularly on PSZ, indeed demonstrate the structure of graphite-like crystalline carbon became more uniform after the shear/friction experiments, i.e. $d_{002}$ became smaller, while $L_c$, $f_a$ and $I_{26}/I_{20}$ became larger (see Fig. 9 and Table 3). Note that our results show $L_a$ became smaller upon shear/friction experiments, which is also observed in coal ranking from low volatile bituminous to semi-anthracite during coalification process (Jiang et al., 2019). This development of molecular structure likely reflects an increase in maturity of the coal samples, which is in good agreement with our observations from Raman spectra. As is known that many Raman parameters are correlated with maturity index (such as vitrinite reflectance (VR) and total fixed carbon), and the correlations are widely reported (Wilkins et al., 2014; Schito et al., 2017; Henry et al., 2018, 2019; Zhang and Li, 2019). Generally, G-FWHM, D-FWHM and $I_D/I_G$ are negatively related to measured %VR, while RBS and saddle index are opposite. To better illustrate the change in coal maturity from our Raman spectra, we plotted two representative correlations between Raman parameters and maturity reported by Henry et al. (2019) in Fig. 11a, i.e. G-FWHM vs. VR and RBS vs. VR. It is clearly seen in Fig. 11a that the apparent differences in Raman parameters (G-FWHM and RBS) between the sheared and unsheared samples show that the coal maturity was improved after the shear/friction experiments. We also note that the petrological and chemical analyses showed our coal sample has a vitrinite reflectance of $0.77 \pm 0.05$ %, which is larger than the values of VR of the unsheared sample S* (~0.6) and S0 (~0.4–0.5) obtained from the correlations shown in Fig. 11a. This suggests it would be difficult to quantitively determine the maturity increase upon the shear/friction process using these correlations between Raman parameters and VR. Nonetheless, these correlations shown in Fig. 11a, we believe, have demonstrated that the Raman parameters can reflect coal maturity change, and once again, the change in Raman parameters accordingly indicates an increase in coal maturity upon the shear/friction experiments.

We now attempt to elucidate the likely mechanisms responsible for the development of molecular structure and maturity of coal during the shear/friction process. It is well known that molecular structure and maturity of coal would be changed under conditions of high temperature and pressures at a geological timescale during metamorphism (Bonijoly et al., 1982; Oberlin, 1984; Buseck and Huang, 1985; Buseck and Beyssac, 2014). Research on pyrolysis of bituminous coal showed that, when the temperature is below ~200 °C, volatilization of small molecules and slight breaking of aliphatic chains would occur to produce only trace amounts of gas (e.g. CO and/or $CH_4$), while above ~300–400 °C, the amount of most functional groups would decrease rapidly with the cracking of C-H bonds and C-C bonds, generating much gas (Öztaş and Yürüm, 2000; Zhao et al., 2007; Niu et al., 2016; Kaneki et al., 2018). Besides, high-resolution TEM, Raman spectroscopy and FTIR analysis performed on the tectonically deformed coals, which often accumulate many shear strains associated with strain energy, suggest that tectonic deformation can improve the ordering of aromatic structure and reduce the content of hydrogen and

oxygen (Ju et al., 2005; Cao et al., 2007; Xu et al., 2014; Pan et al., 2017; Song et al., 2018). Therefore, returning to our experiments, under the PT conditions employed (i.e. 25–45 MPa effective normal stress and 100 ℃), we believe, the

445 molecular structure of bituminous coal is very unlikely to be changed over hours without the shear/friction process (Hou et al., 2014; Xu et al., 2014). Moreover, the slip rates (0.1–100 μm s$^{-1}$) employed in our experiments means the development of molecular structure and maturity due to sample-scale frictional heating effects seen in high-velocity friction experiments can be eliminated (Rice, 2006; Di Toro et al., 2011; Aharonov and Scholz, 2018). In addition, we infer that flash heating mechanisms reported by Kaneki and Hirono (2019) may play a little or minor role, as the structural and Raman parameters

are both insensitive to applied effective normal stress. As a result, we infer that the improvement of molecular structure and maturity observed in our experiments is caused by shear deformation (i.e. strain localization in the shear bands) associated with strain energy (Hou et al., 2017). Similarly, an improvement of the molecular structure of anthracite coal was also observed in the creep compaction experiments at axial strain rates of $1.3\times10^{-6}$–$1.3\times10^{-4}$ s$^{-1}$, performed at a confining pressure of 500 MPa and temperatures of 300–600 ℃ (Ross et al. (1990; 1991). Ross and co-authors (1990; 1991), based on TEM

observations on the deformed samples, found that stacks of aromatic layers in coal became more ordered and was progressively aligned in the plane of flattening during deformation, i.e. the organization of crystalline carbon was significantly improved during the deformation, but not enough to form graphite. Aside from creep compaction experiments, Bustin et al. (1995a, 1995b) also performed simple shear experiments at a confining pressure of 0.8 or 1 GPa on bituminous and anthracite coal at temperatures of 400–900 ℃ employing shear strain rates of $1\times10^{-5}$–$1\times10^{-6}$ s$^{-1}$, as opposed to our shear

strain rates of ~$1\times10^{-4}$–$1\times10^{-1}$ s$^{-1}$ and PT conditions. They found, only for the sheared samples, graphite was formed from bituminous coal initially at 800 ℃ and commonly at 900 ℃ while from anthracite coal initially at 600 ℃ and commonly at 900 ℃. Similar findings were also reported by Wilks et al. (1993). They all suggest that the improvement of molecular structure in coal observed at relative low PT conditions and at lab time scales was caused by the shear strain that can largely lower the activation energy for graphitization process of coal (Ross and Bustin, 1990; Ross et al., 1991; Wilks et al., 1993;

Bustin et al., 1995a, 1995b). By contrast, Nakamura et al. (2015) found that graphite in natural fault zones releases strain energy by micro/nano-scale delamination and size reduction as a function of the degree of deformation, leading to amorphization. This shear-induced (or strain-induced) graphitization process as a mechanism, from a point view of microstructure, may mainly consist of a) promoting preferred orientation and the rearrangement of aromatic structure systems; b) favoring the motion and modulation of structural defects to produce a highly ordered graphite structure (e.g.

Beyssac et al., 2002; Wang et al., 2019).

**4.2 Mechanisms causing slip weakening**

All of our experiments have shown a marked slip weakening behaviour of coal at the initial displacement of ~1–2 mm (Fig. 3), i.e. friction coefficient decreased from a peak value approaching ~0.5 to a near steady state value around only 0.3. Similar slip weakening behaviour was also observed on simulated coal-shale gouges when coal content ≥50 vol% (Liu et al., 2020)

and synthetic graphite gouge (Kirilova et al. 2018) under similar experimental conditions. Kirilova et al. (2018) inferred that

the slip weakening behaviour seen in the synthetic graphite gouge could be related to the degree of order of crystal sheet structure in the shear zone (Morrow et al., 2000; Rutter et al., 2013). Recall that mineral content in our coal samples is ~5% only and EDS data shows little mineral content in the WDZ and PSZ zones. This all suggests the presence of minerals should play little role in controlling this significant slip-weakening behaviour. Moreover, the temperature and sliding rates employed in our experiments, together with our XRD results, mean that lubrication effects (Di Toro et al., 2011) due to sample-scale frictional heating and graphitization effects widely seen in high-velocity friction experiments (e.g. Oohashi et al., 2011) can be eliminated. Besides, this marked slip-weakening behaviour was observed in both samples under vacuum dry and samples exposed to DI water at 15 MPa. This suggests that local overpressure effects of DI pore water as a mechanism (e.g. Faulkner et al., 2018) can be eliminated. On the other hand, our microstructure observations on the deformed gouges shown in Fig. 6 and Fig. 7 indicate the development of R-, B-, Y- shear bands. Following Logan et al. (1992), strain localization in shear bands may be one of the mechanisms responsible for this slip-weakening behaviour of coal. Importantly, our XRD and Raman analysis on the PSZ of the sheared samples suggest the crystallinity was improved in the shear bands upon strain localization associated with strain energy (also see Sect.4.1). This shear-enhanced crystallinity was also observed in natural (deformed) carbonaceous materials in fault zone (Kuo et al., 2018; Wang et al., 2019), which likely results in a reduction in frictional strength (Ruan and Bhushan, 1994; Morrow et al., 2000; Moore and Lockner, 2004). Significantly, in high maturity materials such as graphite, structural disorder is enhanced by shear deformation (Nakamura et al., 2015; Kirilova et al., 2018; Kaneki and Hirono, 2019). As reported by Oohashi et al. (2011), the peak and steady-state friction of amorphous carbon at low slip rates yields ~0.5 while graphite yields ~0.1, suggesting the molecular structure of carbon may play a significant role in controlling frictional properties. Recall that our XRD showed that no graphite was formed after the friction experiments. We, therefore, infer the improvement of crystallinity in coal may dominate the steady-state friction of coal approaching ~0.3, which lies in between the steady-state friction of amorphous carbon and graphite. As a result, this all suggests that the marked slip-weakening behaviour of coal could largely be attributed to both the development of R-, B- and Y- shear bands and an increase in crystallinity of coal in the shear bands upon strain localization. However, we also note that little amounts of gas (e.g. CO and/or $CH_4$) might be produced, in potential, upon volatilization of small molecules and slight breaking of aliphatic chains during shear deformation. It still yet remains unknown whether this process occurred in our experiments and how this process would influence the slip-weakening behaviour. By contrast, Fan and Liu (2019) found no slip weakening behaviour in friction experiments on pre-cut bituminous coal blocks employing shear rates of 1-10 $\mu$m s$^{-1}$ to a total shear displacement of ~7 mm, at an effective normal stress of 2 MPa. This difference between the findings reported by Fan and Liu (2019) and our results suggests that strain energy may play a significant role in enhancing coal crystallinity and in reducing frictional strength. Unfortunately, we cannot directly investigate the effect of strain energy on the development of crystal structure in the present experiments, as we have insufficient experimental data. To further test the hypothesis that strain energy accumulated in the shear bands enhances crystallinity of coal, we will initiate another study involving high-resolution BIB-SEM, Raman and XRD observations on coal gouges subjected to well controlled shear displacements, such as 1, 2, 4, 5 mm and beyond, at low and high normal effective stresses.

## 4.3 Velocity dependence of friction

Our velocity stepping (VS) experiments, from an RSF point of view, indicate sliding velocity and pore fluid play roles in rate dependence of friction of coal (see Fig. 4b). In an attempt to better illustrate these effects, we plot the (*a-b*) values obtained from the experiments S2–S7 against up-step sliding velocity shown in Fig. 12. We note that almost all wet coal samples S3-S7 (except S5) show positive *a-b* values in the velocity stepping of 1-10 μm s$^{-1}$. Similar velocity strengthening behaviour of pre-cut bituminous coal exposed to various fluids (helium, carbon dioxide, water, and moisturized methane) at an effective normal stress of 2 MPa in sliding velocities of 1-10 μm s$^{-1}$ was also reported by Fan and Liu (2019). Careful inspection of Fig. 12 shows a clear, systematic tendency for a transition from velocity strengthening to velocity weakening, regardless of the experimental conditions employed in this study. Interestingly, the transition occurs at a slip rate of ~10 μm s$^{-1}$ in samples S3–S7 exposed to DI water at 15 MPa, but may occur at a slip rate of ~1 μm s$^{-1}$ in the sample S2 under vacuum dry condition. As this transition from velocity strengthening to velocity weakening is seen both in vacuum dry samples and in samples exposed to pore water at 15 MPa, we infer that the local fluid overpressure effects (e.g. Faulkner et al., 2018) may play a little or minor role. On the other hand, we note that, unlike mineral gouges, coal exhibits marked stress-strain-sorption behaviour when exposed to water or/and gas (Liu et al., 2016, 2018). This behaviour leads to swelling/shrinkage strains that strongly depend on chemical activity (pressure) of the adsorbing fluid as well as on the Terzaghi effective stress supported by the solid grain framework. Such effects could conceivably result in competition between compaction and dilatation during shearing of coal gouge, leading to a complex rate dependence of friction similar to that produced by competition between dilatant granular flow and compaction by pressure solution seen in mineral gouges (Niemeijer and Spiers, 2007; Chen and Spiers, 2016). In the following, we attempt to determine whether the above mechanism proposed by Chen, Niemeijer and Spiers can logically explain the observed transitions.

We first focus on the coal sample under vacuum dry condition, in which the transition from velocity strengthening to weakening occurred at a sliding velocity of ~1 μm s$^{-1}$. This, following Niemeijer and Spiers (2007), suggests that for the vacuum dry coal, the rate of compaction was similar to the dilatation rate during shear deformation at the imposed sliding velocities of ~1–100 μm s$^{-1}$. This may be reasonable, because our XRD results, as well as the experimental results reported by the literature (Ross and Bustin, 1990; Ross et al., 1991), suggest that shear deformation (strain) changes the coal molecular structure so that stacks of aromatic layers become more uniformly packed, thus enhancing the rate of compaction to the same order as the rate of the dilatation process at sliding velocities >~1 μm s$^{-1}$. Indeed, as seen in oedometer-type (1D) compaction creep experiments performed by Liu et al. (2018), using coal powders from the same source as the present study, wet coal powder exhibits much larger compaction strains than vacuum dry coal, as well as an increase in compaction rate of 1–2 orders of magnitude. This is broadly consistent with the compaction strains estimated from the thickness change of the present wet versus vacuum dry samples, measured before versus after the experiments. This is clearly seen in Table 1 which shows 19-30% compaction strain in wet experiments S3-S8 compared with 10-19% compaction in dry samples S1 and S2.

According to Liu et al. (2018), pore water enhances compaction of coal powder through a) permanent time-dependent compaction (creep) and b) the thermodynamic effect of a stress-driven reduction in water sorption capacity and an associated reduction in swelling with respect to dry material. In this case, enhanced compaction rates in the wet coal samples compared to vacuum dry samples may dominate the competition against dilation at sliding velocities of ~1–10 μm s$^{-1}$, thus promoting velocity strengthening. As a result, our present study suggests the microphysical model developed by Chen, Niemeijer and Spiers (Niemeijer and Spiers, 2007; Chen and Spiers, 2016) may logically explain the observed rate dependence of friction. However, other mechanisms cannot be completely eliminated. More research is needed for a better understanding of such behaviour, for example, experiments employing a broader range of slip rates or investigation of the effects of the compaction rate of coal exposed to different pore fluids.

Besides, we note, from our previous direct shear experiments performed on simulated coal-shale fault gouges under similar PT conditions (see Liu et al., 2020), that the 50:50 volume fraction coal-shale mixtures exhibited a tendency for a transition from velocity weakening to velocity strengthening at a sliding velocity of ~10 μm s$^{-1}$, as opposed to pure coal samples reported here. This strongly suggests that the presence of mineral phases may play a significant role in rate dependence of friction of coal-rich faults.

### 4.4 Implications for frictional strength and seismic potential of coal-rich faults

Our experiments demonstrate that bituminous coal exhibits significant slip-weakening at an initial displacement of ~1–2 mm, when a sliding velocity of 1 μm s$^{-1}$ is imposed under both dry and wet conditions, at a temperature of 100 ℃ and effective normal stress of 25–45 MPa. Post-test analysis suggests this marked slip-weakening was caused by the development of strain localization in shear bands, accompanied by the improvement of the organization of molecular structure and accordingly of coal maturity. Similar slip weakening behaviour associated with coal-rich shear bands was also observed on simulated coal-shale gouges when coal content ≥ 50 vol%, under similar PT conditions at sliding velocities of 0.1–100 μm s$^{-1}$ (Liu et al., 2020). This suggests that this marked slip weakening of coal may play a role in promoting unstable slip of coal or coal-rich faults, i.e. facilitating accelerating fault slip and earthquake nucleation in coal-rich fault segments, given appropriately low stiffness of the surrounding rock volume. However, the shear-unload-reshear experiments performed by Liu et al. (2020) on 50:50 volume fraction coal-shale mixtures suggest that significant slip-weakening occurs only in previously unsheared material in which coal-rich shear bands have yet to develop (Logan et al., 1992; Marone, 1998). This suggests that slip-weakening may not be prone to occur on coal-rich faults at the low slip velocities associated with rupture nucleation, as previous tectonic displacements have potentially exceeded the 1–2 mm slip weakening distance observed in our experiments. This is highly uncertain, however, since diagenetic processes in faults during periods of no-slip may rework the gouge fabric, so that renewed slip weakening may occur upon reactivation. Alternatively, marked slip-weakening may occur in coal seams when new fractures are produced upon failure of the intact material, caused by underground coal mining or (enhanced) coalbed methane recovery operations. In addition, frictional slip of coal-rock interface may also play an important role in the

stability of coal-bearing faults (Wang et al., 2014; Li et al., 2015b). On the other hand, our SHS experiment shows minor healing (restrengthening) effects of bituminous coal, with transient peak healing in friction increases with the logarithm of hold time (s) at a linearized rate of only ~0.006. This healing rate would become even smaller when adding 50 vol% shale (see Liu et al. 2020). Post-healing slip weakening effects are correspondingly minor, compared with healing rates typically
measured in quartz or carbonate gouges (Nakatani and Scholz, 2004; Chen et al., 2015). We accordingly expect a minor or little effect of healing on frictional strength of coal-coal or coal-rich faults, even after geological periods of healing, which again points to very limited scope for slip weakening and seismogenic rupture nucleation in the case of fault reactivation. Besides slip weakening effects, we note that the near steady-state frictional strength of coal yields ~0.3, which is insensitive to pore fluids and effective normal stresses employed in this study. This suggests the mechanical heterogeneity caused by
weak coal-coal and coal-smear patches may also play a role in promoting instability of faults (Tembe et al., 2010; Kohli and Zoback, 2013; Buijze et al., 2017; Kaneki and Hirono, 2019).

Our VS experiments demonstrate a transition from velocity strengthening to velocity weakening at a slip rate of ~10 μm s$^{-1}$ for wet coal at 15 MPa pore water pressure, while this occurs at ~1 μm s$^{-1}$ for vacuum dry coal. We inferred this transition
was caused by competition between dilatant granular flow and compaction by stress-strain-sorption behaviour of coal. We note that this rate-dependent friction of coal, observed at slow slip velocities, may be changed by adding other minerals (e.g. coal-shale mixtures reported by Liu et al., 2020), as the dilatation/compaction processes and accordingly their competition will be changed. Therefore, whether coal-coal or smeared out coal-rich fault patches exhibit slip-weakening behaviour and potentially cause seismogenic rupture nucleation (Scholz, 2019), remains an open question for a given fault.

**5 Conclusions**

This paper has investigated frictional properties of simulated bituminous coal gouges under (near) in situ conditions of 100 °C and 25–45 MPa effective normal stresses, employing sliding velocities of 0.1–100 μm s$^{-1}$. We determined the rate dependence of friction following RSF theory. Also, frictional healing effect of coal were investigated by performing a single slide-hold-slide experiment with hold durations in the range 300 to 30000 s. Microstructures of the deformed gouges were
investigated and the development of the molecular structure of coal was analyzed using XRD and Raman spectroscopy. The likely mechanisms controlling the frictional behaviour were then discussed. The main findings are summarized as follows:

1. All simulated coal gouges showed marked slip weakening behaviour, from a peak friction coefficient approaching ~0.5 to a near steady-state value around only ~0.3, regardless of the applied effective normal stress (25–45 MPa) or the presence of water (at 15 MPa pore pressure) versus vacuum dry conditions.
2. Microstructural observations, performed on the sheared samples using optical microscopy and SEM, showed that all coal gouges have developed a principal boundary slip zone with marked stack-layer structure, and a weakly

deformed zone. A single sample (S5) showed clear R- and Y- shear bands. This suggests strain localization in the shear bands of coal gouge.

3. The crystal structure of the coal samples was investigated using XRD. Our results showed that a) no graphite was formed due to the shear/friction processes; b) following shear deformation, the interlayer spacing of graphite-like structure ($d_{002}$) decreased from ~3.57 to 3.47–3.53, while the stacked height ($L_c$) and aromaticity ($f_a$) increased from ~13 Å to ~20 Å and from 0.44–0.49 to 0.53–0.67, respectively. This suggests an improvement in crystallinity of the sheared coal samples, which may be caused by strain localization associated with strain energy developed in the shear bands.

4. Raman spectroscopy showed that a) D and G bands were present in all samples at ~1360 cm$^{-1}$ and ~1600 cm$^{-1}$, respectively; b) both D and G bands observed in the sheared samples are more narrow than those observed in the unsheared samples; c) G-FWHM values decreased from 91.3 cm$^{-1}$ to 71.7–79.7 cm$^{-1}$ following shear deformation, while RBS and SI values increased from 227.7 cm$^{-1}$ to 239.3–256.7 cm$^{-1}$ and from 3.27 to 3.65–4.26, respectively. This all indicates an increase of coal maturity, which is in good agreement with our XRD results.

5. Based on above findings of 2–4, we suggest that the marked slip-weakening behaviour of coal could largely be attributed to the development of R-, B- and Y- shear bands, i.e. strain localization in shear bands, accompanied by the improvement of crystallinity and maturity of coal.

6. From an RSF point of view, our VS experiments showed a transition from velocity strengthening to velocity weakening at slip velocity of ~1 μm s$^{-1}$ in the coal sample under vacuum dry condition, but at ~10 μm s$^{-1}$ in coal samples exposed to DI water at 15 MPa pore pressure. This may be dominated by competition between dilatant granular flow and compaction enhanced by the presence of water.

7. The single SHS experiment revealed transient peak healing (restrengthening) which increased log-linearly with hold time at a rate of ~0.006, demonstrating minor time-dependent healing.

8. Our findings, together with our previous research on frictional properties of coal-shale mixtures, suggest that the rather marked slip weakening of coal may not be prone to occur on pre-existing coal-rich faults with a well-developed, localized internal structure, at least for low slip velocities associated with rupture nucleation, but may occur in coal seams when failure or new fractures are produced by underground coal mining or (enhanced) coalbed methane recovery. Such behaviour may lead to accelerating slip and the associated possible seismic hazard. Based on our single SHS experiment, healing is expected to play only a minor role in the frictional strength evolution of coal-rich faults, even for geological time scales, which again points to very limited scope for slip weakening and seismogenic rupture nucleation in the case of fault reactivation. Besides, the rate dependence of friction observed in coal gouges may change due to the addition of other minerals (e.g. coal-shale mixtures), which suggests that the seismogenic potential of coal-bearing faults, via slip-weakening behaviour, remains an open question for a given fault.

**Data availability**

The landing page for all original data is https://public.yoda.uu.nl/geo/UU01/48I5DA.html, alongside doi:10.24416/UU01-48I5DA.

**Author contribution**

Investigation and research were performed by all of the authors. Specifically, Caiyuan Fan performed XRD and Raman experiments, processed all data and wrote original draft under supervision of Dr. Jinfeng Liu; Jinfeng Liu, with the help of Luuk B. Hunfeld, performed friction experiments and produced metadata; Jinfeng Liu and Prof. Christopher J. Spiers formulated the ideas and research goals of this manuscript; Jinfeng Liu, Luuk B. Hunfeld and Christopher J. Spiers conducted critical review and revisions.

**Competing interests**

The authors declare that they have no conflict of interest.

**Acknowledgement**

This research was funded by the National Natural Science Foundation of China (NSFC Project No. 41802230). Dr. Jianye Chen is thanked for discussions and HPT Lab technicians, Gert Kastelein and Floris van Oort, are thanked for their superb technical support. Luuk B. Hunfeld and part of the experimental work conducted at Utrecht were supported through the research program on induced seismicity in the Groningen Gas Field funded by the field operator, the Nederlandse Aardolie Maatschappij (NAM).

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

**Table 1. List of experiments, experimental conditions and key mechanical data. VS = velocity stepping, SHS = slide-hold-slide. Note that all experiments reported here were performed at ~100 °C. $\sigma_n$ and $P_f$ represent the confining pressures and pore fluid pressure employed in the experiments. $\mu_{peak}$ represents the peak friction coefficient obtained at 0.5–0.75 mm shear displacement, $\mu_{ss1}$ and $\mu_{ss2}$ represent the near steady-state friction coefficient obtained at ~2.2 mm and ~5.7 mm shear displacement respectively. $D_{tot}$ represents the total shear displacement. $t_0$ and $t$ represent the thickness of the gouge layer measured before and after the experiments, respectively. $\tau_{ss2}$ and $\varepsilon$ represent the steady-state shear stress and shear strain measured at the shear displacement of ~5.7 mm and the latter is defined as engineering shear strain, i.e. $\varepsilon = d/t_0$, where $d$ is the shear displacement. Here, $\varepsilon$ equals 5.7 divided by initial thickness $t_0$.**

| Exp./ Sam. | $\sigma_n$ [MPa] | $P_f$ [MPa] | $\mu_{peak}$ [-] | $\mu_{ss1}$ [-] | $\mu_{ss2}$ [-] | $D_{tot}$ [mm] | $V$ [µm s⁻¹] | $t_0$ [mm] | $t$ [mm] | $\tau_{ss2}$ [MPa] | $\varepsilon$ [-] |
|---|---|---|---|---|---|---|---|---|---|---|---|
| S* | * | * | | | | | | | | | |
| S0 | 20 | * | | | | | | | | | |
| **VS** | | | | | | | | | | | |
| S1 | 40 | 0 | 0.494 | 0.327 | 0.295 | 6.201 | 0.1–100 | 1.03 | 0.93 | 11.73 | 5.56 |
| S2 | 40 | 0 | 0.465 | 0.278 | 0.244 | 5.563 | 0.1–100 | 0.92 | 0.75 | 9.62 | 6.16 |
| S3 | 40 | 15 | 0.524 | 0.338 | 0.282 | 5.602 | 0.1–100 | 1.00 | 0.75 | 6.87 | 5.70 |
| S4 | 50 | 15 | N/A | 0.279 | 0.253 | 6.183 | 0.1–100 | 1.18 | 0.83 | 8.78 | 4.85 |
| S5 | 55 | 15 | 0.441 | 0.258 | 0.228 | 5.766 | 0.1–100 | 0.90 | 0.65 | 8.91 | 6.33 |
| S6 | 55 | 15 | 0.485 | 0.293 | 0.256 | 6.010 | 0.1–100 | 0.90 | 0.73 | 10.14 | 6.33 |
| S7 | 60 | 15 | 0.454 | 0.273 | 0.245 | 5.750 | 0.1–100 | 1.00 | 0.80 | 10.85 | 5.70 |
| **SHS** | | | | | | | | | | | |
| S8 | 55 | 15 | 0.460 | 0.275 | 0.244 | 6.057 | 1 | 1.00 | 0.80 | 9.64 | 5.70 |

S* represents the starting, loose coal powders without the pre-compaction process.
S0 represents the compacted coal gouge layer only, i.e. without the shear deformation.
S* and S0 are used as the control samples for XRD and Raman tests, in an attempt to determine the effects of shear deformation on the molecular structure of carbon in coal.
N/A: This value is missing.

**Table 2. Summary of RSF data for all velocity-stepping experiments reported in this paper.**

| Sam./Steps | $V_0$ [µm s⁻¹] | $V$ [µm s⁻¹] | $a-b$ [-] | $a$ [-] | $b$ [-] | $D_c$ [mm] | $\eta$ [mm⁻¹] |
|---|---|---|---|---|---|---|---|
| **S1** | | | | | | | |
| V_step1 | 0.1 | 1 | N/A | N/A | N/A | N/A | -0.010 |
| V_step2 | 1 | 10 | -0.0055 | 0.0051 | 0.0094 | 0.0074 | -0.016 |
| V_step3 | 10 | 100 | N/A | N/A | N/A | N/A | N/A |
| **S2** | | | | | | | |
| V_step1 | 0.1 | 1 | 0.0001 | 0.0030 | 0.0029 | 0.0029 | -0.009 |
| V_step2 | 1 | 10 | -0.0018 | 0.0028 | 0.0047 | 0.0120 | -0.020 |
| V_step3 | 10 | 100 | -0.0047 | 0.0629 | 0.0676 | 0.0440 | -0.036 |
| **S3** | | | | | | | |
| V_step1 | 0.1 | 1 | N/A | N/A | N/A | N/A | -0.008 |
| V_step2 | 1 | 10 | 0.0010 | 0.0034 | 0.0024 | 0.0032 | -0.035 |
| V_step3 | 10 | 100 | -0.0042 | 0.0637 | 0.0679 | 0.0370 | -0.071 |

**S4**

| | | | | | | | |
|---|---|---|---|---|---|---|---|
| V_step1 | 0.1 | 1 | 0.0061 | 0.0128 | 0.0067 | 0.0014 | N/A |
| V_step2 | 1 | 10 | 0.0012 | 0.0043 | 0.0031 | 0.0101 | -0.016 |
| V_step3 | 10 | 100 | -0.0012 | 0.3271 | 0.3283 | 0.0578 | -0.025 |

**S5**

| | | | | | | | |
|---|---|---|---|---|---|---|---|
| V_step1 | 0.1 | 1 | 0.0052 | 0.0327 | 0.0276 | 0.0014 | -0.007 |
| V_step2 | 1 | 10 | -0.0019 | 0.0057 | 0.0076 | 0.0364 | -0.017 |
| V_step3 | 10 | 100 | -0.0049 | 0.1008 | 0.1057 | 0.0436 | -0.022 |

**S6**

| | | | | | | | |
|---|---|---|---|---|---|---|---|
| V_step1 | 0.1 | 1 | 0.0135 | 0.0470 | 0.0336 | 0.0071 | -0.011 |
| V_step2 | 1 | 10 | 0.0009 | 0.0039 | 0.0030 | 0.0134 | -0.025 |
| V_step3 | 10 | 100 | -0.0008 | 0.0091 | 0.0099 | 0.0234 | -0.023 |

**S7**

| | | | | | | | |
|---|---|---|---|---|---|---|---|
| V_step1 | 0.1 | 1 | 0.0014 | 0.0047 | 0.0033 | 0.0067 | -0.005 |
| V_step2 | 1 | 10 | 0.0001 | 0.0029 | 0.0028 | 0.0143 | -0.019 |
| V_step3 | 10 | 100 | -0.0054 | 0.4209 | 0.4264 | 0.0586 | -0.023 |

N/A: This value cannot be obtained as the fitting quality is low.


**Table 3. Coal crystal structure parameters determined from XRD profiles.**

| Sam. | | $2\theta_{002}$ [°] | $2\theta_{10}$ [°] | $\beta_{002}$ [°] | $\beta_{10}$ [°] | $d_{002}$ [Å] | $L_c$ [Å] | $L_a$ [Å] | $I_{26}/I_{20}$ | $f_a$ |
|---|---|---|---|---|---|---|---|---|---|---|
| | S* | 24.81 | 43.59 | 5.81 | 8.94 | 3.58 | 14.26 | 20.22 | 1.29 | 0.44 |
| | | 24.93 | 43.59 | 5.48 | 8.38 | ±0.01 | ±0.59 | ±0.92 | ±0.10 | ±0.04 |
| | S0 | 25.02 | 43.72 | 6.10 | 8.13 | 3.56 | 12.87 | 21.46 | 1.21 | 0.49 |
| | | 24.94 | 43.80 | 6.41 | 8.18 | ±0.01 | ±0.45 | ±0.09 | ±0.03 | ±0.03 |
| P S Z | S1 | 25.57 | 44.73 | 3.88 | 11.79 | 3.48 | 20.76 | 14.90 | 1.72 | 0.56 |
| | S3 | 25.25 | 43.96 | 5.28 | 9.13 | 3.52 | 15.25 | 19.18 | 1.33 | 0.53 |
| | S4 | 25.62 | 44.17 | 3.77 | 11.04 | 3.47 | 21.37 | 15.88 | 1.76 | 0.56 |
| | S5 | 25.46 | 44.30 | 3.92 | 10.82 | 3.49 | 20.22 | 15.97 | 2.23 | 0.67 |
| | | 25.51 | 44.30 | 4.05 | 11.15 | ±0.00 | ±0.46 | ±0.34 | ±0.00 | ±0.00 |
| | S7 | 25.57 | 44.36 | 4.06 | 10.58 | 3.48 | 19.84 | 16.58 | 1.96 | 0.61 |
| | S8 | 25.21 | 43.82 | 5.18 | 9.10 | 3.53 | 15.54 | 19.24 | 1.69 | 0.61 |
| W D Z | S1 | 25.22 | 44.17 | 4.61 | 10.96 | 3.53 | 17.46 | 15.99 | 2.11 | 0.64 |
| | S3 | 25.55 | 43.80 | 4.51 | 9.52 | 3.48 | 17.86 | 18.39 | 1.51 | 0.51 |
| | S4 | 25.34 | 44.11 | 4.49 | 8.97 | 3.51 | 17.93 | 19.54 | 2.29 | 0.63 |
| | S5 | 25.40 | 43.97 | 4.55 | 10.77 | 3.50 | 17.70 | 16.26 | 1.58 | 0.54 |
| | S7 | 25.28 | 44.21 | 4.65 | 7.74 | 3.52 | 17.31 | 22.65 | 1.98 | 0.60 |
| | S8 | 25.54 | 43.67 | 4.27 | 10.27 | 3.48 | 18.86 | 17.04 | 1.16 | 0.43 |

PSZ and WDZ represent the principal boundary slip zone and weakly deformed zone retrieved from the sheared samples.

Note that the parameter for Sample S*, S0 and S5 are mean values, associated with standard deviation, obtained from the reproducible tests.

**Table 4. Parameters obtained from normalized Raman spectra using the method reported by Henry et al. (2018).**

| Sam. | | S* | S4* | S0 | S1 | S2 | S3 | S4 | S5 | S6 | S7 | S8 |
|---|---|---|---|---|---|---|---|---|---|---|---|---|
| D Position | $[cm^{-1}]$ | 1368 | 1354 | 1369 | 1359 | 1364 | 1361 | 1362 | 1361 | 1352 | 1348 | 1347 |
| Stdev | | / | 10.1 | 7.0 | 8.5 | 1.5 | 7.5 | 8.7 | 2.5 | 8.4 | 2.9 | 3.0 |
| D-FWHM | $[cm^{-1}]$ | N/A | N/A | N/A | 209.4 | N/A | N/A | N/A | N/A | N/A | 217.3 | 201.3 |
| Stdev | | / | N/A | N/A | 7.7 | N/A | N/A | N/A | N/A | N/A | 4.6 | 10.1 |
| G position | $[cm^{-1}]$ | 1601 | 1601 | 1597 | 1603 | 1601 | 1601 | 1602 | 1601 | 1603 | 1601 | 1604 |
| Stdev | | / | 0.0 | 3.5 | 1.5 | 0.0 | 0.0 | 1.2 | 0.6 | 1.7 | 0.6 | 0.6 |
| G-FWHM | $[cm^{-1}]$ | 84.0 | 78.0 | 91.3 | 73.9 | 77.7 | 79.0 | 79.7 | 78.0 | 77.0 | 77.3 | 71.7 |
| Stdev | | / | 4.6 | 4.5 | 4.4 | 0.6 | 2.6 | 4.0 | 1.0 | 6.3 | 0.6 | 3.5 |
| RBS | $[cm^{-1}]$ | 233.0 | 246.7 | 227.7 | 244.5 | 237.3 | 240.0 | 239.7 | 239.3 | 250.5 | 252.3 | 256.7 |
| Stdev | | / | 10.1 | 10.3 | 9.8 | 1.5 | 7.5 | 9.2 | 3.1 | 10.1 | 3.2 | 3.1 |
| $I_D/I_G$ | / | 0.540 | 0.548 | 0.543 | 0.553 | 0.555 | 0.561 | 0.543 | 0.553 | 0.553 | 0.552 | 0.551 |
| Stdev | | / | 0.024 | 0.004 | 0.009 | 0.006 | 0.014 | 0.006 | 0.013 | 0.019 | 0.011 | 0.003 |
| SI | / | 3.70 | 3.84 | 3.27 | 4.26 | 3.65 | 3.78 | 3.76 | 3.65 | 3.75 | 3.79 | 4.08 |
| Stdev | | / | 0.65 | 0.34 | 0.32 | 0.10 | 0.33 | 0.09 | 0.08 | 0.30 | 0.10 | 0.13 |

S4* represents the weakly deformed zone of sample S4.

N/A: This value cannot be obtained due to the limitations of the processing method described in Sect.2.5.3.


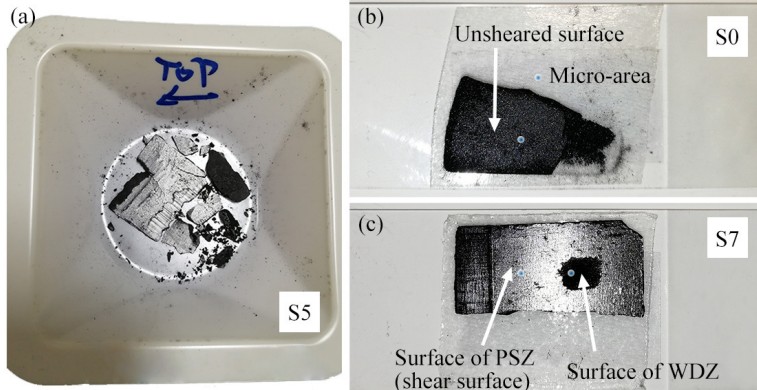

**Figure 1: Post-treatment of the recovered fragments used for microstructural observation, XRD and Raman spectroscopy. (a) Stored loose fragments. (b) The free surface of the unsheared sample S0 that was glued on a glass slide. (c) The slip surface of the sample S7, showing the surface of the principal slip zone (PSZ) and weakly deformed zone (WDZ).**


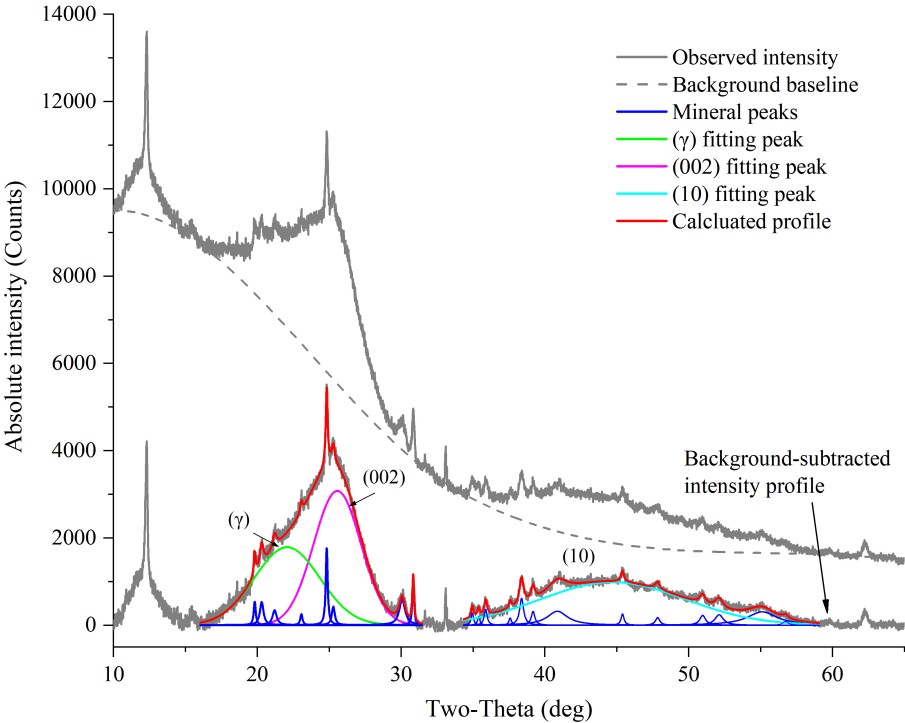

**Figure 2: Fitting curves in an X-ray diffraction pattern for the sample S1.**


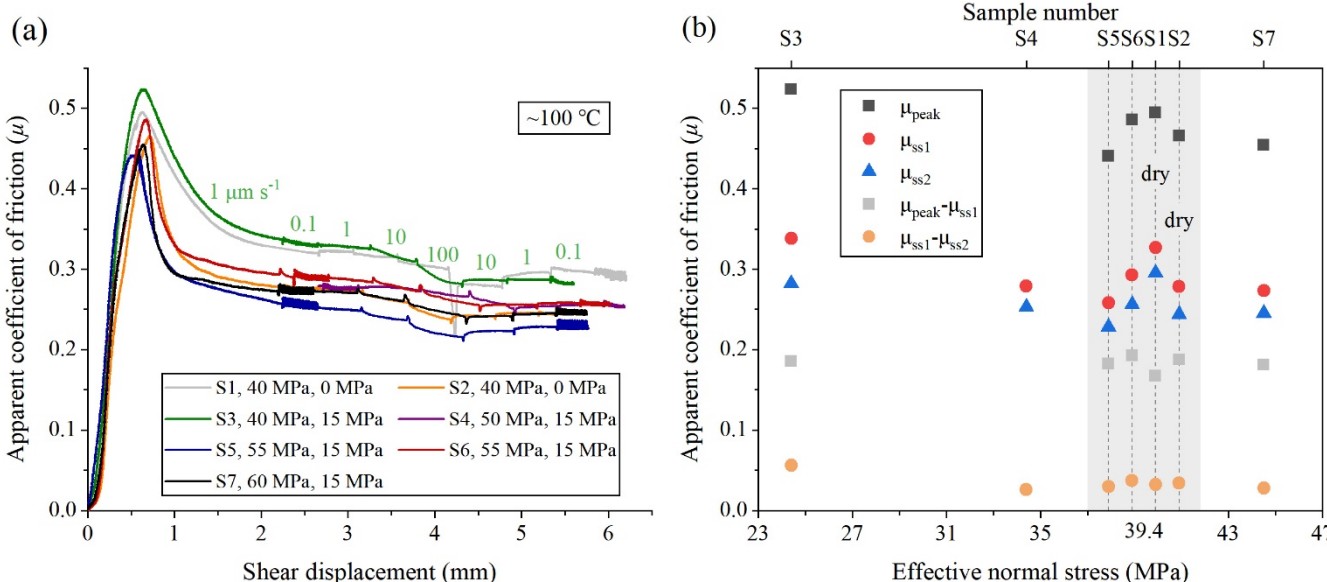

**Figure 3:** Frictional properties of the samples S1–S7 obtained from the velocity stepping experiments. **(a)** Apparent coefficient of friction ($\mu$) against shear displacement. **(b)** Apparent coefficient of friction ($\mu$) against **effective normal stress. Note that data offset is given in the grey area, i.e. the data plotted on the vertical dashed lines were all obtained at ~40 MPa effective normal stress but are horizontally offset here for readability.**

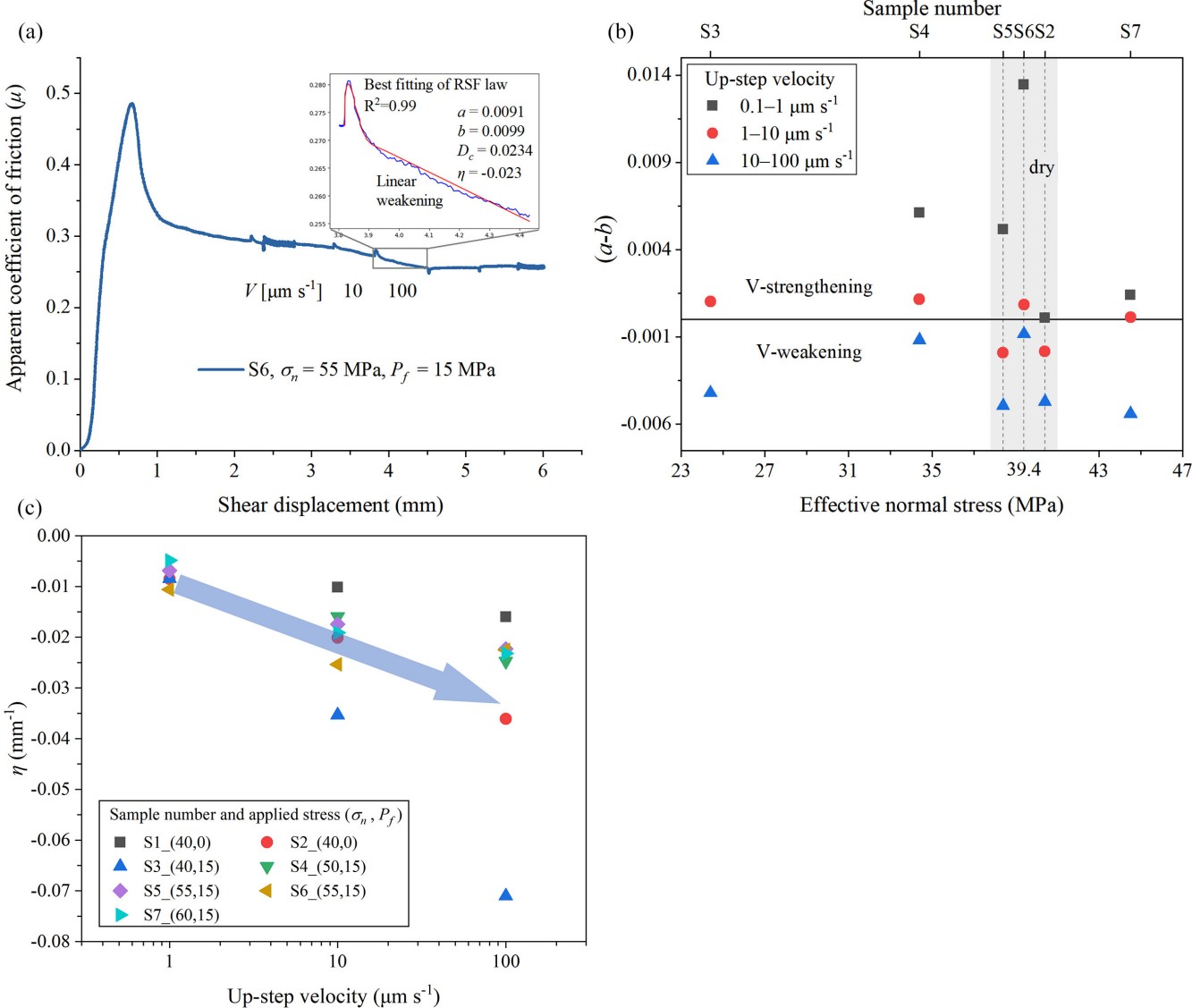

**Figure 4: (a)** Results of experiment S6, illustrating the best fitting of a full RSF law to the experimental data obtained at **the velocity step from 10 to 100 µm s⁻¹**. **(b)** ($a$-$b$) values, obtained from upward velocity steps using a full RSF fit, **versus effective normal stress at a linear scale. Note that data offset is given in the grey area, i.e. the data plotted on the vertical dashed lines were all obtained at ~40 MPa effective normal stress but are horizontally offset here for readability. (c) The slope $\eta$ of linear slip weakening trend versus up-step velocity in velocity stepping.**

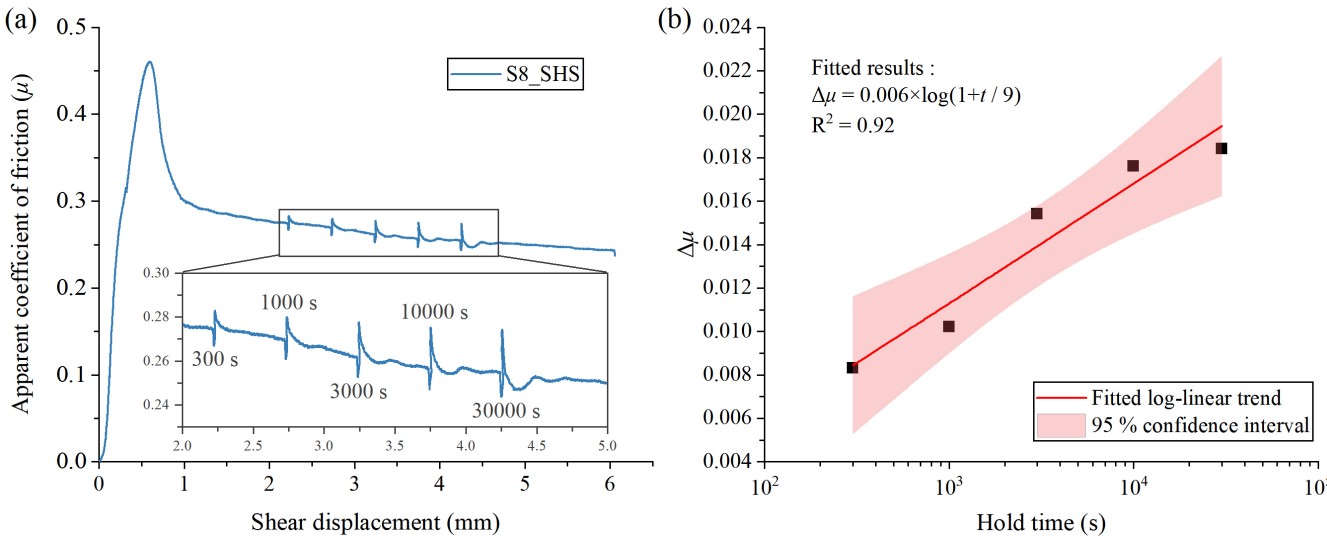


**Figure 5: Slide-hold-slide experimental data for sample S8 tested at a pore fluid pressure of 15 MPa and a confining pressure of 55 MPa. (a) Friction coefficient versus shear displacement, showing slide-hold-slide testing sequence. (b) Transient peak healing or post-hold frictional restrengthening plotted as a function of the logarithm of hold time. The black solid squares represent the experimental data derived from (a).**


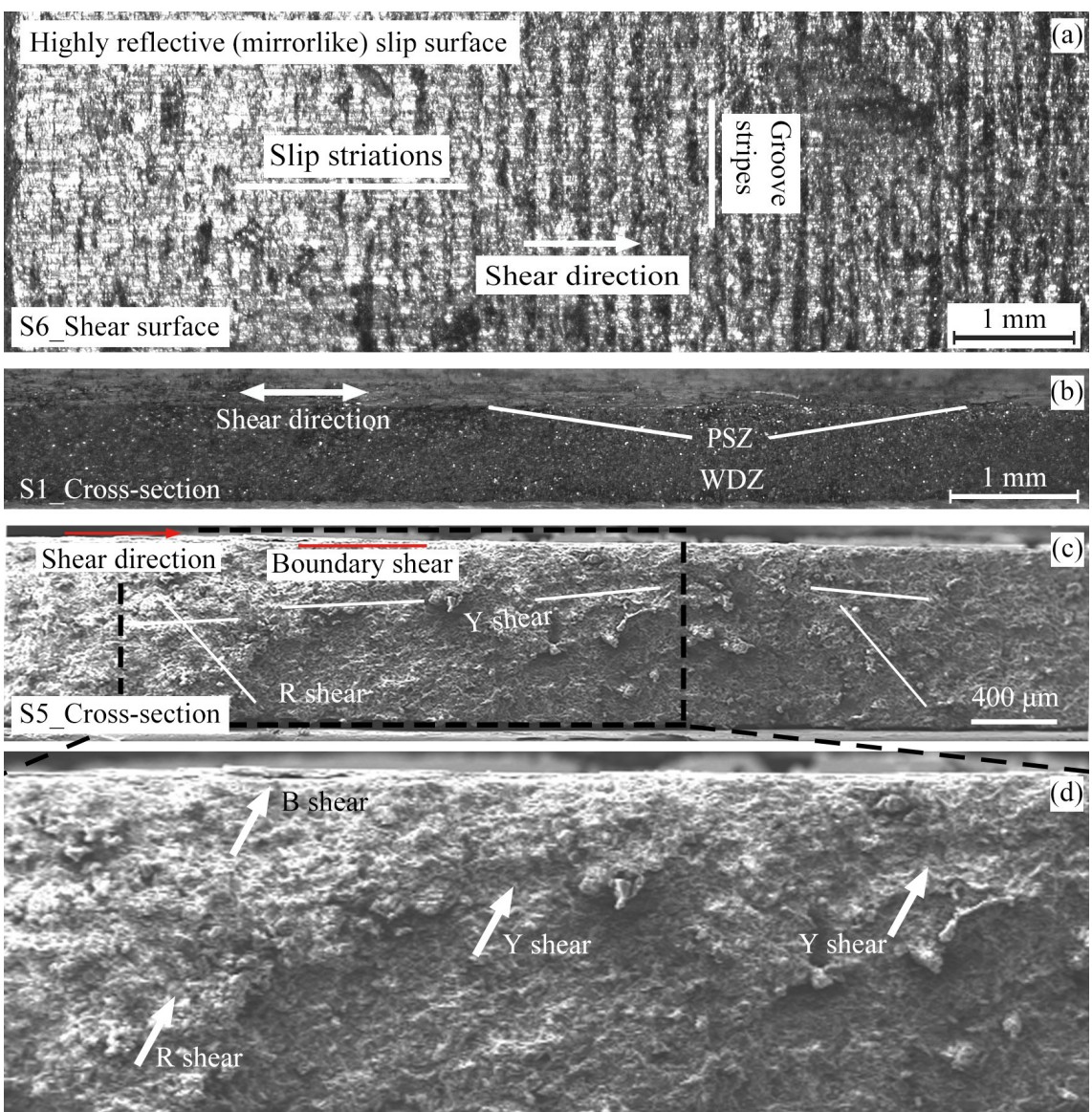

**Figure 6: Microstructure of samples S1, S5 and S6 after shear deformation. (a) and (b) were imaged using an optical microscope in a reflected light mode, while (c) was imaged using SEM in secondary electron mode. (a) The shear surface of sample S6, showing a highly reflective (mirrorlike) slip surface on the left hand side of the image (below corresponding label). (b) The cross-section of sample S1 in an orientation parallel to the shear direction, indicating a principal slip zone (PSZ) and a weakly deformed zone (WDZ). (c) The cross-section of sample S5 in an orientation parallel to the shear direction, showing the development of R shear, boundary and Y shear bands. (d) The magnification of the region marked in (c).**


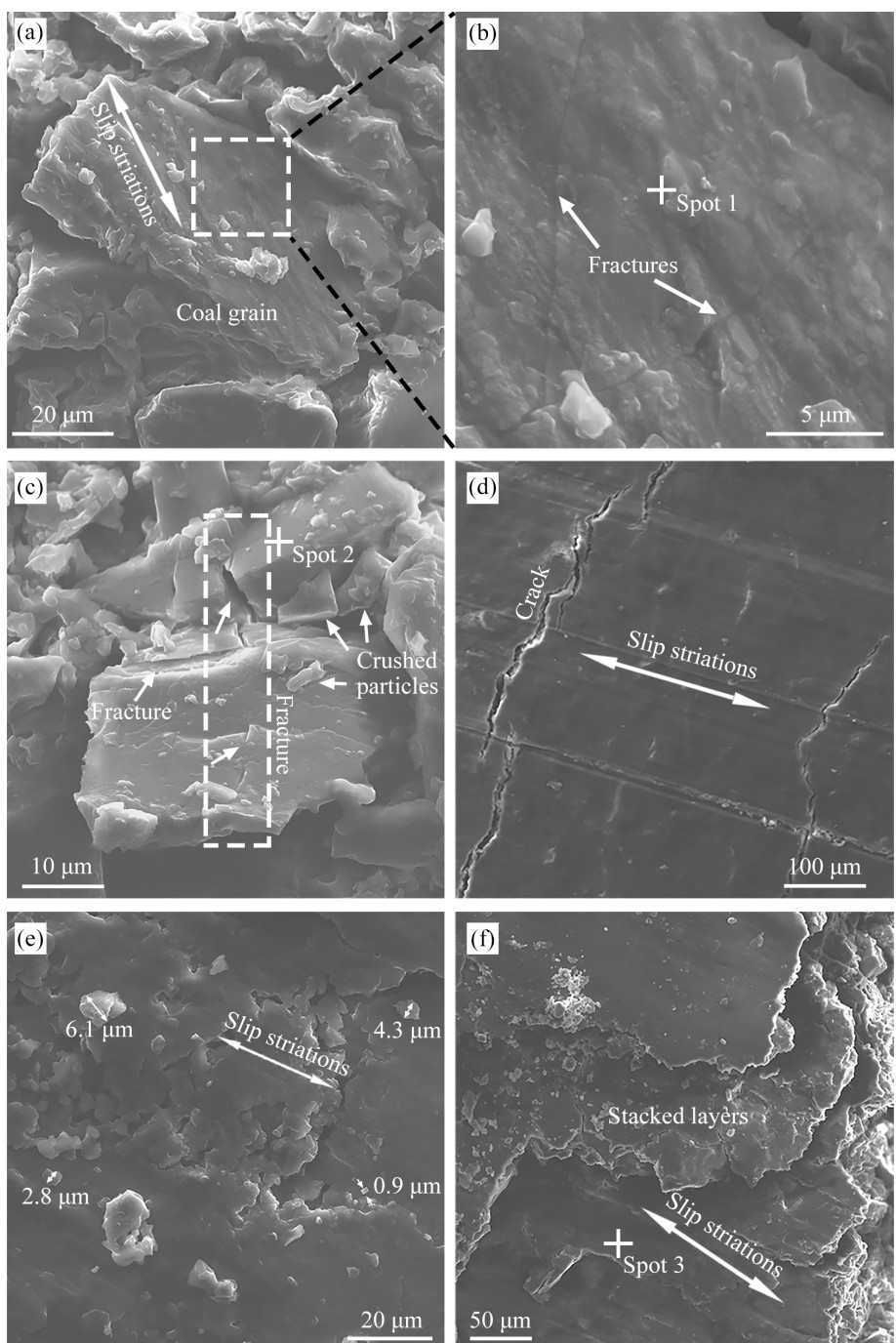

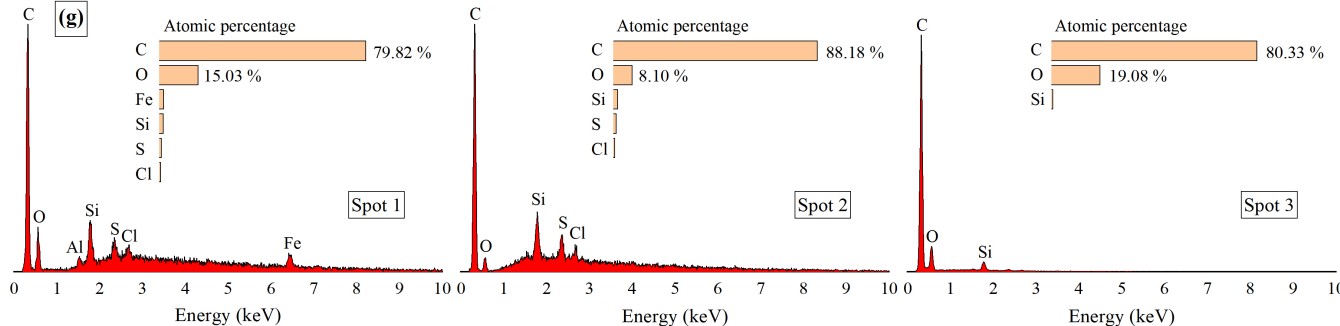


**Figure 7: SEM secondary electron images of the samples S4 (a, b, c, f) and S5 (d, e) after the direct shear experiments and EDS data (g) for the representative spots. (a) Randomly oriented coal grain with slip striations in the WDZ, located in the cross-section of S4. (b) The fractures inside the coal grain shown in (a). (c) Fractures inside coal grains to form crushed particles. (d) The central region of the shear surface of sample S5, showing cracks and slide stripes. (e) Small coal particles (<~10 μm) in the broken**
**edge of the shear surface. (f) Remarkable layered structure at the margins of the slip zone, likely reflecting an interaction between PSZ and WDZ, and the role of PSZ during the friction process. (g) EDS data for three spots located in S4, showing the elemental composition of the spots.**

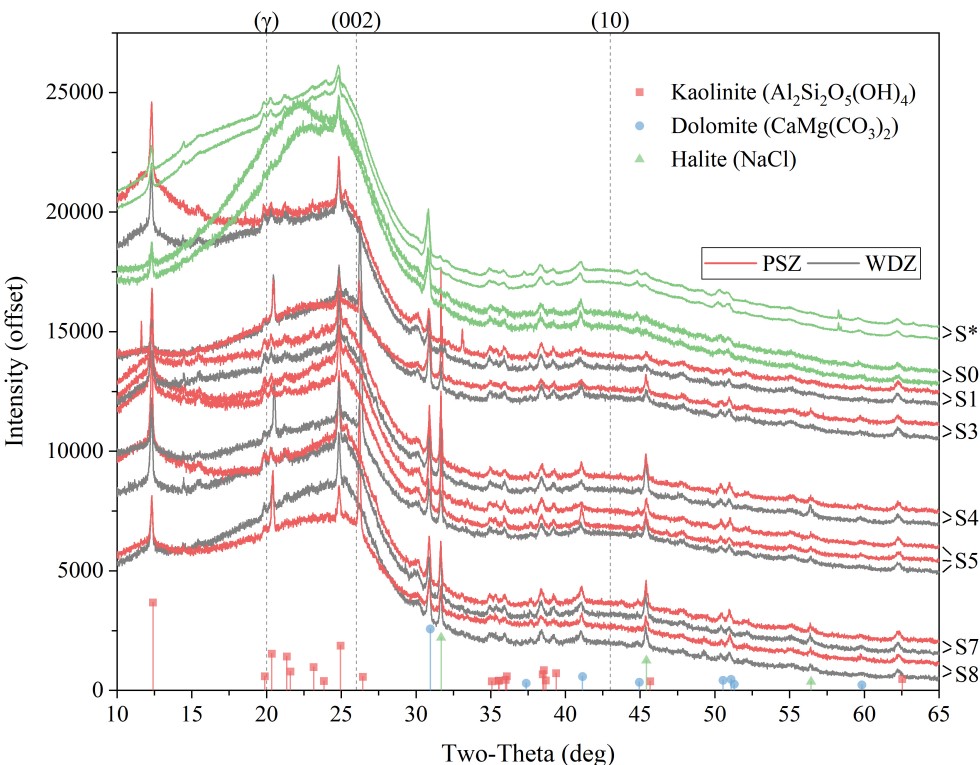


**Figure 8: X-ray diffractograms for starting coal powder S*, pre-compacted coal (unsheared) sample S0, and sheared samples S1–S8 except S2 and S6. The minor peaks are marked by possible minerals using points. The red and black lines represent the XRD profile for samples retrieved from the principal boundary slip zone (PSZ) and the weakly deformed zone (WDZ). Note that the intensity for sample S* was reduced by ten times to a comparable scale.**

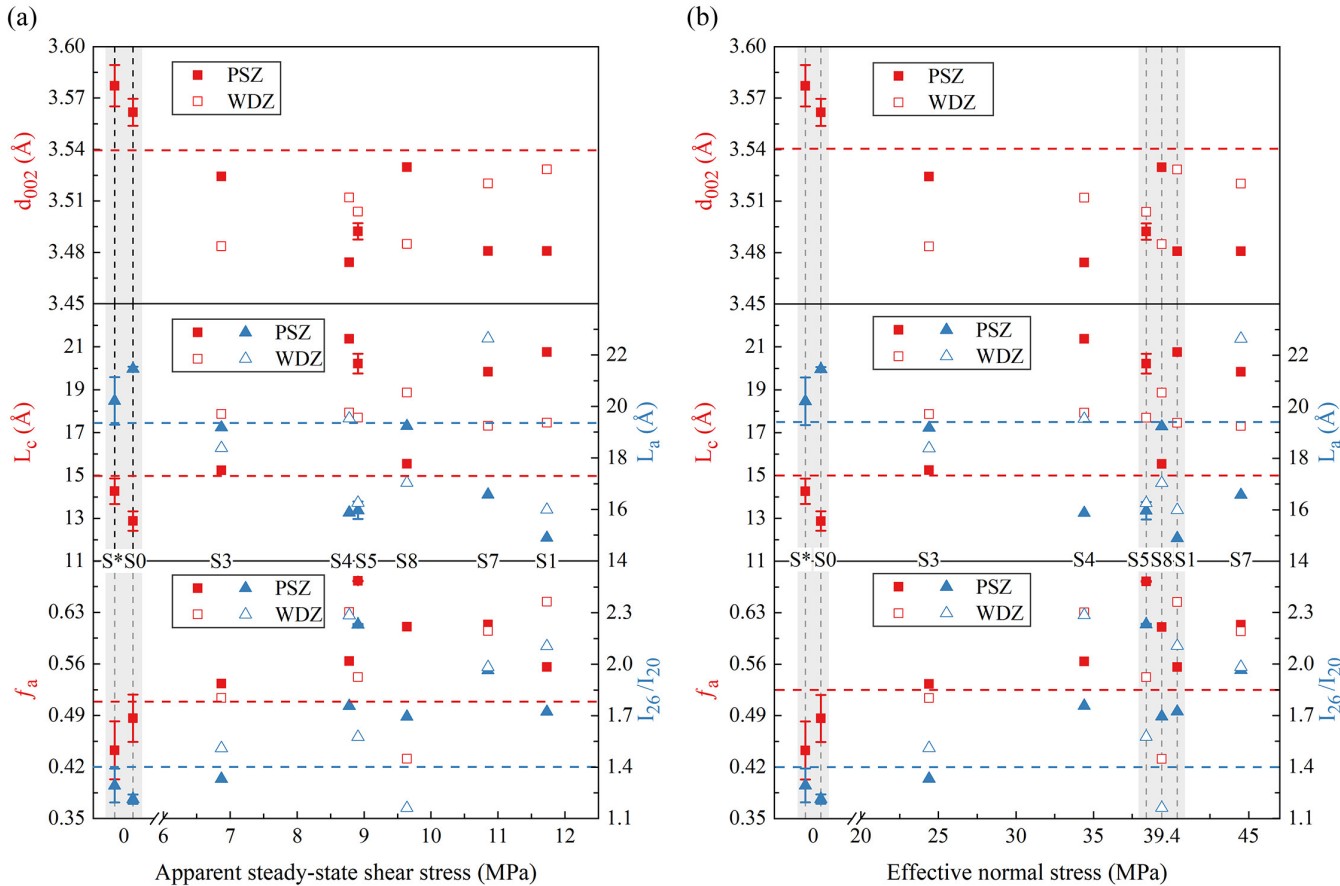

**Figure 9: Representative crystal structure parameters obtained from the samples S\*–S8 versus apparent steady-state shear stress measured at a shear displacement of ~5.7 mm and effective normal stress with corresponding sample number in (a) and (b) respectively. Data offset is given in the grey area, i.e. the data plotted on the vertical dashed lines were obtained at 0 MPa (non-sheared samples) or ~40 MPa effective normal stress but are horizontally offset here for readability. Solid and hollow squares or triangles represent the values for the PSZ and WDZ retrieved from the sheared samples, respectively. The error bars for samples S\*, S0 and S5 are the standard deviations.**

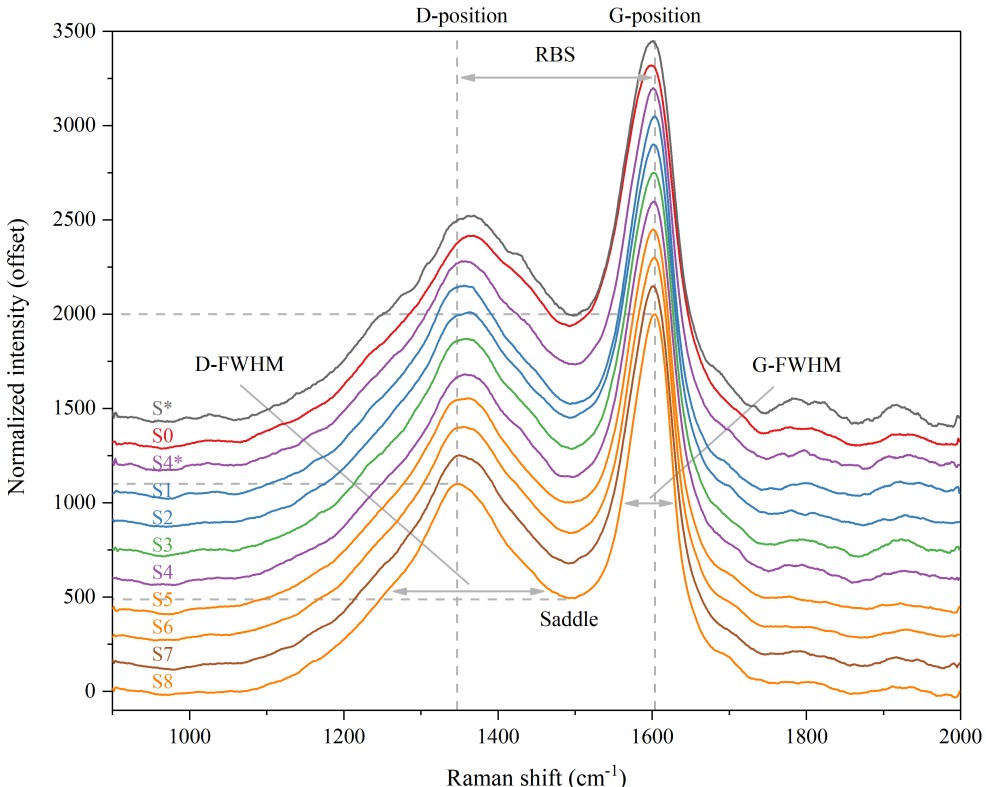

**Figure 10: The processed Raman spectra of coal samples, showing the spectra differences between unsheared samples (S* and S0) and sheared samples (S1–S8). Note that different colours mean different experimental conditions and S4* represents the weakly deformed zone of the sample S4. Some important parameters were marked according to Henry et al. (2018).**

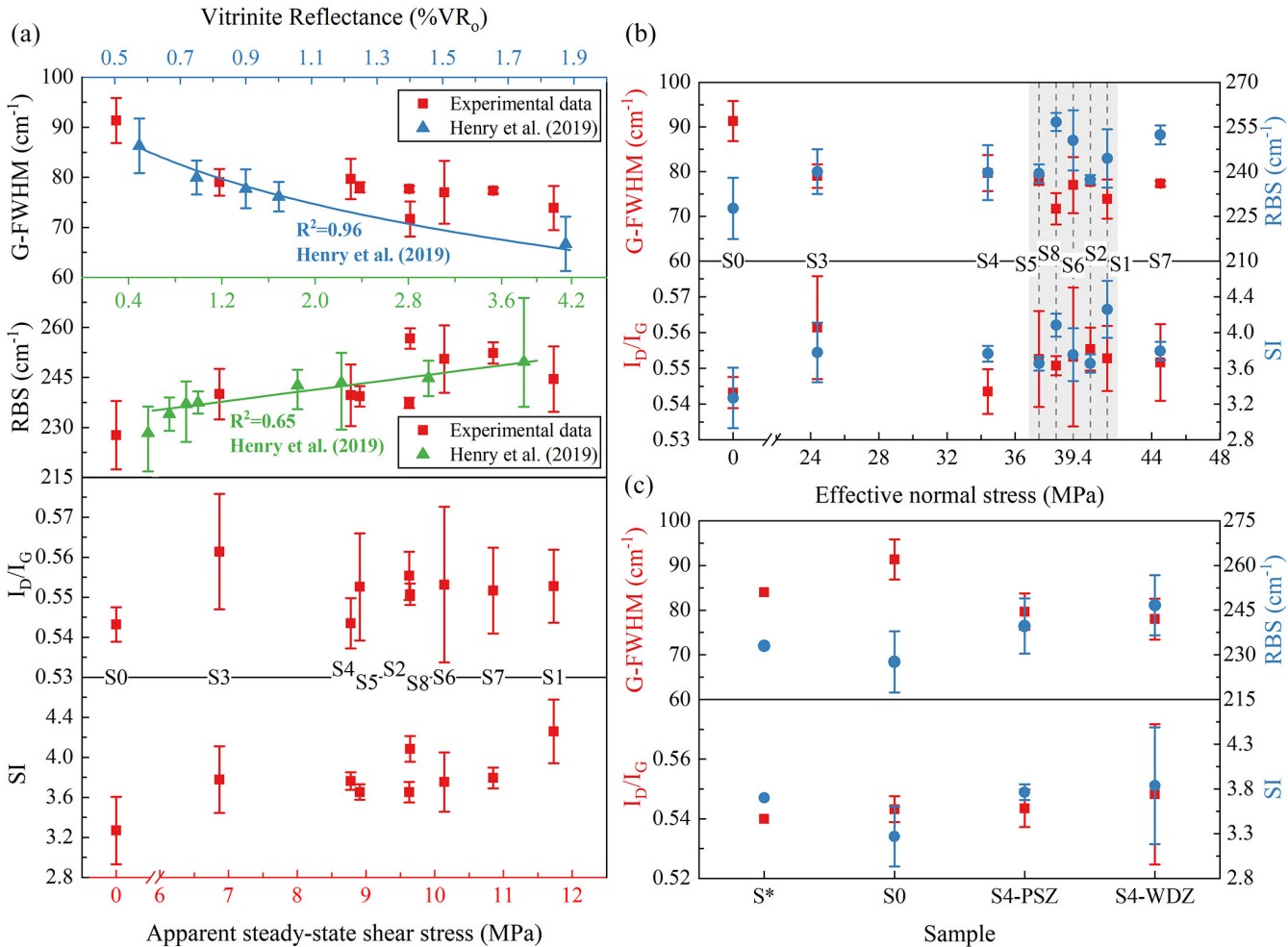


**Figure 11: Representative Raman parameters as a function of apparent steady-state shear stress and effective normal stress plotted in (a) and (b) respectively. The difference of Raman parameters between S*, S0, S4-PSZ, and S4-WDZ is shown in (c). In figure (b), data offset is given in the grey area, i.e. the data plotted on the vertical dashed lines were obtained at ~40 MPa effective normal stress but are horizontally offset here for readability.** Note that the relation of G-FWHM vs. vitrinite reflectance and RBS vs. vitrinite reflectance plotted in (a) were obtained from Henry et al. (2019). The error bars are the standard deviations.


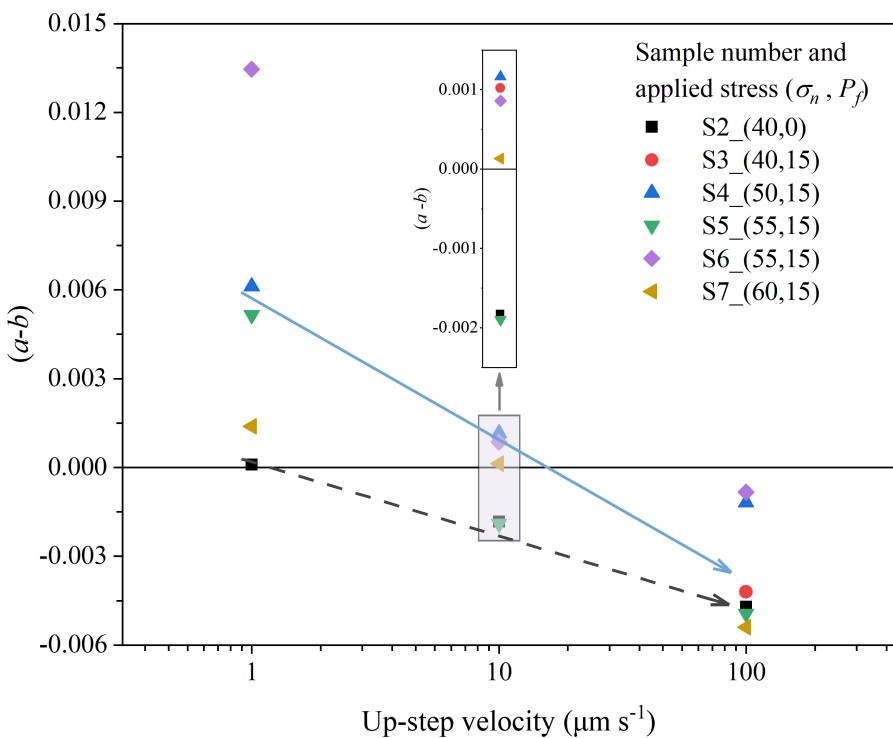

**Figure 12:** (*a-b*) values obtained from the experiments S2–S7 versus up-step velocity, showing a transition from velocity strengthening to velocity weakening.