# Peer review of "Frictional slip weakening and shear-enhanced crystallinity in simulated coal fault gouges at slow slip rates"

_Solid Earth, 2020_

## Referee Comment (RC1) · Anonymous Referee #1 · 9 Apr 2020

This is an unusual format for a review - for me. But I will supply comments as in a usual review.

The authors have completed a thorough study of the frictional characteristics of organic materials - in particular focusing on observations and in linking these observations with a mechanistic understanding - an din this case with a particular focus on chryslaine products and chemical transformation.

The work is thorough, has important conclusions - velocity sealing at high shear velocities and due to the creating of microcrystalline products - and is comprehensively but compactly presented.

---

## Referee Comment (RC2) · Oohashi Kiyokazu (Referee) · 4 May 2020

The manuscript entitled "Frictional slip weakening and shear-enhanced crystallinity in simulated coal fault gouges at subseismic slip rates" describes frictional properties of coal at slip rates of 0.1-100 $\mu$m/s and crystallographic characteristics before and after the friction experiments. The authors report marked slip weakening behavior during the experiments and attribute the behavior to a shear-enhanced crystallographic development of coal based on XRD and Raman analysis. The manuscript is basically well written and their results seem sound. Hence I recommend accepting it with minor revisions.

[Figure]

Title: "Subseismic slip rates" makes me imagine much higher velocities of mm/s to cm/s. I would rather use "slow slip rates".

Line 15: whether vacuum dry flooded → whether vacuum dry/flooded ?

Line 35: very low friction strength → very low frictional strength

Line 92: such as friction strength → such as frictional strength

Line 221: "$\gamma$-band, 002-band and 10-band" I think the majority of readers are not familiar with $\gamma$-band. Please explain briefly, or cite an adequate reference.

Lines 256-260: Here the authors briefly explain types of data which are summarized in Table 1 and 2, however, I think it is not necessarily to mention them as a first sentence of chapter. Please consider to incorporate it in 3.1.1 and 3.2.2. Also, I'm wondering why the authors define two types of steady-state friction coefficient ($\mu$ss1 and $\mu$ss2), in other words, why all the mechanical data have a slight decrease in frictional strength at about 4 mm in displacement. I ask you to mention this behavior and reason for it (if possible) in the description of overall frictional behavior (Lines 263-265).

Line 265: "though this trend is not significant" Line 276: "(a-b) values may not be sensitive to effective normal stress" Here the authors mention the relationships between frictional strength (or (a-b) value) and effective normal stress. However, in Fig.3b and Fig.4b, apparent friction coefficients are plotted for each experiment (not plotted against effective normal stress (or (a-b) value)), and the trend is not visualized. Although the value of applied stress is shown individually in the figures, it is quite difficult to see any scientific meaning (relationship between X and Y axis) of the plot. The same problem can be seen in Fig. 9 and Fig. 11. I would plot apparent friction coefficients (or (a-b) values) against effective normal stress.

Line 279: "while other samples tested with DI pore water at a pressure of 15 MPa show velocity strengthening." There is an exception (S5) that exhibits velocity-weakening behavior for wet experiments. Please explain correctly.

Line 288: "the onset of the slip surface" Unclear meaning.

Line 299: Here the authors explain the strain localization during the experiment; slip on R-shear surface takes place in the early stage of deformation, and then deformation mode changes to slip on boundary and Y-shear bands in the late stage of experiment. However, authors document that R- and Y- shear bands were only observed in the sample S5 due to flat condition of the sample surface. My question is how do you know the time sequence of formation of R-, Y-, and boundary shear bands based on your observations?

Line 325: Citation of Fig. 9c should appear after Fig. 9b.

Lines 352-356: Delete first 4 lines of "Discussion" section to avoid repetition.

Line 490: "enhanced compaction rates" Do you have any direct evidences for enhanced compaction of wet sample? (e.g., thickness of the layer)

Figures

Fig. 2: I'm wondering why the background (especially for the range of 10-30 degree in two-theta) of observed intensity is so high? Is this due to a specimen holder or diffractometer used in this study?

Fig. 3b and 4b: I would plot apparent friction coefficients (or (a-b) values) against effective normal stress.

Fig. 6c: I can hardly see R- and Y-shear bands in the photograph. Please provide a magnified photograph.

Fig. 7: Description/citation of Fig. 7g does not appear anywhere in the main text.

Fig. 9: I think the horizontal axis should be an effective normal stress and/or apparent steady-state shear stress with linear scale, to indicate your data in a more scientific way. Each crystal structure parameter versus an effective normal stress and apparent steady-state shear stress should be illustrated separately using different symbol. The

[Figure]

information of shear strain does not seem important because the values are almost identical.

Fig. 11a and 11b: Here I also suggest the horizontal axis should be an effective normal stress and/or apparent steady-state shear stress. Also please consider plotting the crystal structure parameters and Raman parameters against frictional work (shear stress*displacement, MJ/m^2 or MJ/kg ) stored in the sample, as you finally conclude that the improvement in crystallinity may be associated with strain energy.

Fig. 12: Please provide "Sample number and applied stress ($\sigma$n, Pf)" in the figure legend.

Kiyokazu Oohashi

––––––––––––––––––––––––––––––––

---

## Editor Comment (EC1) · Jianye Chen (Editor) · 22 May 2020

Dear Autors,

We have received two reviews which agree on the high importance, quanlity, and time-liness as well as clear writing of your manuscrit. In agreement with the reviewers, I cer-taintly recommend publication after addressing the comments and formatting issues, which are considerred to be both minor.

Two addtionsl comments which may be helpful: 1. I agree that there is rare data available on the friction strength, stability and healing of carbonaceuous materials. The

authors could dig a bit on this and give a short review on or comparison with previous studies.

2. In your microstructures, you have marked many 'fractures'. If I understand correctly, you are analyzing fragments not thin section. I sugget being careful about the fractures. Some may form when you splited the sample.

Jianye Chen

---

## Author Comment (AC1) · 19 Jun 2020

Thanks for your nice words.

———————————————

---

## Author Comment (AC2) · 19 Jun 2020

Thanks for your comments. Please see the detailed replies in the Supplement pdf. The figures were also revised following the suggestions.

Please also note the supplement to this comment: https://se.copernicus.org/preprints/se-2020-43/se-2020-43-AC2-supplement.pdf

[Figure]

[Figure]

Figure 3: Frictional properties of the samples S1–S7 obtained from the velocity stepping experiments. (a) Apparent coefficient of friction (μ) against shear displacement. (b) Apparent coefficient of friction (μ) against effective normal stress. Note that data offset is given in the grey area, i.e. the data plotted on the vertical dashed lines were all obtained at ~40 MPa effective normal stress but are horizontally offset here for readability.

**Fig. 1.** Figure 3: Frictional properties of the samples S1–S7 obtained from the velocity stepping experiments.

[Figure]

(a)

Best fitting of RSF law
R²=0.99
$a$ = 0.0091
$b$ = 0.0099
$D_c$ = 0.0234
$\eta$ = -0.023

Linear
weakening

$V$ [μm s⁻¹]   10   100

—— S6, $\sigma_n$ = 55 MPa, $P_f$ = 15 MPa

Apparent coefficient of friction ($\mu$)

Shear displacement (mm)

(b)

Sample number
S3        S4   S5S6S2   S7

Up-step velocity
■ 0.1–1 μm s⁻¹
● 1–10 μm s⁻¹
▲ 10–100 μm s⁻¹

dry

$(a$-$b)$

V-strengthening

V-weakening

Effective normal stress (MPa)

(c)

$\eta$ (mm⁻¹)

Sample number and applied stress ($\sigma_n$, $P_f$)
■ S1_(40,0)        ● S2_(40,0)
▲ S3_(40,15)      ▼ S4_(50,15)
◆ S5_(55,15)      ◄ S6_(55,15)
► S7_(60,15)

Up-step velocity (μm s⁻¹)

Figure 4: (a) Results of experiment S6, illustrating the best fitting of a full RSF law to the experimental data obtained at the velocity step from 10 to 100 μm s⁻¹. (b) (*a*-*b*) values, obtained from upward velocity steps using a full RSF fit, versus effective normal stress at a linear scale. Note that data offset is given in the grey area, i.e. the data plotted on the vertical dashed lines were all obtained at ~40 MPa effective normal stress but are horizontally offset here for readability. (c) The slope $\eta$ of linear slip weakening trend versus up-step velocity in velocity stepping.

**Fig. 2.** Figure 4 in the manuscript

[Figure]

Figure 6: Microstructure of samples S1, S5 and S6 after shear deformation. (a) and (b) were imaged using an optical microscope in a reflected light mode, while (c) was imaged using SEM in secondary electron mode. (a) The shear surface of sample S6, showing a highly reflective (mirrorlike) slip surface on the left hand side of the image (below corresponding label). (b) The cross-section of sample S1 in an orientation parallel to the shear direction, indicating a principal slip zone (PSZ) and a weakly deformed zone (WDZ). (c) The cross-section of sample S5 in an orientation parallel to the shear direction, showing the development of R shear, boundary and Y shear bands. (d) The magnification of the region marked in (c).

**Fig. 3.** Figure 6: Microstructure of samples S1, S5 and S6 after shear deformation.

(a)

[Figure]

**Figure 9: Representative crystal structure parameters obtained from the samples S*–S8 versus apparent steady-state shear stress measured at a shear displacement of ~5.7 mm and effective normal stress with corresponding sample number in (a) and (b) respectively. Data offset is given in the grey area, i.e. the data plotted on the vertical dashed lines were obtained at 0 MPa (non-sheared samples) or ~40 MPa effective normal stress but are horizontally offset here for readability. Solid and hollow squares or triangles represent the values for the PSZ and WDZ retrieved from the sheared samples, respectively. The error bars for samples S*, S0 and S5 are the standard deviations.**

**Fig. 4.** Figure 9 in the manuscript

[Figure]

Figure 11: Representative Raman parameters as a function of apparent steady-state shear stress and effective normal stress plotted in (a) and (b) respectively. The difference of Raman parameters between S*, S0, S4-PSZ, and S4-WDZ is shown in (c). In figure (b), data offset is given in the grey area, i.e. the data plotted on the vertical dashed lines were obtained at ~40 MPa effective normal stress but are horizontally offset here for readability. Note that the relation of G-FWHM vs. vitrinite reflectance and RBS vs. vitrinite reflectance plotted in (a) were obtained from Henry et al. (2019). The error bars are the standard deviations.

**Fig. 5.** Figure 11 in the manuscript

**Supplement:**

Reviewer 2

The manuscript entitled "Frictional slip weakening and shear-enhanced crystallinity in simulated coal fault gouges at subseismic slip rates" describes frictional properties of coal at slip rates of 0.1-100 μm/s and crystallographic characteristics before and after the friction experiments. The authors report marked slip weakening behavior during the experiments and attribute the behavior to a shear-enhanced crystallographic development of coal based on XRD and Raman analysis. The manuscript is basically well written and their results seem sound. Hence I recommend accepting it with minor revisions.

**Detailed reply to the comments by Reviewer 2:**

1.    Title: "Subseismic slip rates" makes me imagine much higher velocities of mm/s to cm/s. I would rather use "slow slip rates".

1.    Reply

Thanks for the suggestion. Indeed, "slow slip rates" is more accurate than "subseismic slip rates" to refer to the range 0.1-100 μm/s. We have changed the "subseismic slip rates" to "slow slip rates".

2.    Line 15: whether vacuum dry flooded → whether vacuum dry/flooded ?
       Line 35: very low friction strength → very low frictional strength
       Line 92: such as friction strength → such as frictional strength

2.    Reply

Thanks for your careful review. We have changed the "whether vacuum dry flooded" to "whether vacuum dry / flooded", and "friction strength" to "frictional strength". Other similar errors in our text have also been corrected.

3.    Line 221: "γ-band, 002-band and 10-band" I think the majority of readers are not familiar with γ-band. Please explain briefly, or cite an adequate reference.

3.    Reply

The γ-band is related to the structure of slightly branched but ring-free saturated hydrocarbons, as opposed to the (002) band which is related to the aromatic layer structure. Yen et al. (1961) performed a comparative experimental study of the γ-band and (002) band, so we added a brief description of it in new lines 244-246 as follows:

In general, the γ-band reflects the structure of ring-free saturated hydrocarbons (see detailed description in Yen et al., 1961), whereas the (002) and (10) bands reflect the ring structure of the aromatic layers of crystalline carbon (Lu et al., 2001).

4.  Lines 256-260: Here the authors briefly explain types of data which are summarized in Table 1 and 2, however, I think it is not necessarily to mention them as a first sentence of chapter. Please consider to incorporate it in 3.1.1 and 3.2.2. Also, I'm wondering why the authors define two types of steady-state friction coefficient (μss1 and μss2), in other words, why all the mechanical data have a slight decrease in frictional strength at about 4 mm in displacement. I ask you to mention this behavior and reason for it (if possible) in the description of overall frictional behavior (Lines 263-265).

4.  Reply

Thanks for your suggestion. We have revised the location of the description of the mechanical data obtained. Regarding the slight decrease in frictional strength with displacement between 2 and 4 mm, we regard it as a decrease occurring between two quasi steady-state friction levels and we use $\mu_{ss1}$ and $\mu_{ss2}$ as the corresponding quasi steady-state values of friction coefficient. The slight slip weakening behaviour seen might be caused by the fact that the area used to calculate the shear stress supported by the sample is the initial surface area of the sample, which does not capture the change of real load-supporting area during the shear experiments. We have added the above point in section 3.1.1, as follows in lines 282-288 (marked in red). We also note that in the interval of 2-4 mm displacement, the frictional strength decreases roughly linearly with displacement, at a slip weakening rate that increases with increasing slip velocity, as shown in Fig. 3a. This linear slip weakening behaviour has also been observed in the velocity-stepping (0.03–100 μm/s) experiments performed on natural fault gouge (37–65% clay minerals, up to 40% quartz + plagioclase and little

calcite) collected from Nankai subduction zone in Japan, and has been put forward as a mechanism for promoting slow earthquakes (see details in Ikari et al., 2013). We added these points as follows in new lines 307-313 of the manuscript. These changes in old lines 255-280 are shown in new lines 279-313, as shown below:

[revised manuscript text omitted]

5. Line 265: "though this trend is not significant" Line 276: "(a-b) values may not be sensitive to effective normal stress" Here the authors mention the relationships between frictional strength (or (a-b) value) and effective normal stress. However, in Fig.3b and Fig.4b, apparent friction coefficients are plotted for each experiment (not plotted against effective normal stress (or (a-b) value)), and the trend is not visualized. Although the value of applied stress is shown individually in the figures, it is quite difficult to see any scientific meaning (relationship between X and Y axis) of the plot. The same problem can be seen in Fig. 9 and Fig. 11. I would plot apparent friction coefficients (or (a-b) values) against effective normal stress.

5.  Reply

Understood! We have replotted Fig. 3b and Fig. 4b to directly show apparent friction coefficient and (*a-b*) against effective normal stress. To avoid overlap of the scattered data obtained at 40 MPa effective normal stress, we offset the data parallel to the horizontal axis as shown in the grey-shaded area of Figures 3b and 4b. A similar procedure has also been applied in Fig. 9 and Fig. 11, see Reply 13.

6.  Line 279: "while other samples tested with DI pore water at a pressure of 15 MPa show velocity strengthening." There is an exception (S5) that exhibits velocity-weakening behavior for wet experiments. Please explain correctly.

6.  Reply

We have revised this description in line 305 as follows:

> while samples tested with DI pore water at a pressure of 15 MPa show velocity strengthening, except for sample S5 which exhibits velocity-weakening.

7.  Line 288: "the onset of the slip surface" Unclear meaning.

7.  Reply

This is indeed not very clear. We have revised the description in line 321 in the revised version as follows:

> indicating a highly reflective (mirrorlike) area located in the left half of Fig. 6a.

8.  Line 299: Here the authors explain the strain localization during the experiment; slip on R-shear surface takes place in the early stage of deformation, and then deformation mode changes to slip on boundary and Y-shear bands in the late stage of experiment. However, authors document that R- and Y- shear bands were only observed in the sample S5 due to flat condition of the sample surface. My question is how do you know the time sequence of formation of R-, Y-, and boundary shear bands based on your observations?

8.    Reply

Good point. Indeed, we agree. We cannot get the real time series of the development of shear bands from this present study and have no evidence for this development sequence. We basically followed the model developed by Logan et al. (1992) for compacted calcite fault gouge samples to presume that our simulated coal fault gouge followed a similar sequence. Of course, to better understand the development sequence of the microstructure for coal gouge in our paper, more research (such as BIB-SEM observation) is needed. To solve this in the present paper, we have deleted the "offending" text in original lines 299-301 in Section 3.2.

9.    Line 325: Citation of Fig. 9c should appear after Fig. 9b.

Lines 352-356: Delete first 4 lines of "Discussion" section to avoid repetition.

9.    Reply

We have revised the description of the result in section 3.3 and deleted the first 4 lines of the original "Discussion" section.

10.   Line 490: "enhanced compaction rates" Do you have any direct evidences for enhanced compaction of wet sample? (e.g., thickness of the layer)

10.   Reply

Good point. Unfortunately, we don't have direct evidence to support this evidence, as we cannot measure the compaction (rate) of the gouge layer directly in the experiments. In addition, the pore fluid volume change data is not accurate enough to allow us to calculate the compaction (rate) during the shear deformation upon leaking effects of the pump. However, according to independent oedometer-type (1D) compaction experiments performed by Liu et al. (2018) on compacted coal powders from the same coal materials, wet coal indeed exhibits much larger compaction strains compared to vacuum dry coal. Furthermore, the thickness of the layer measured before and after the experiments listed in Table 1, showed that the estimated compaction strains of 19-30% for wet experiments S3-S8 is larger than the 10-19% measures for dry tests S1 and S2. We modified the lines 537-545 as follows, to state the above points clearly:

Indeed, as seen in oedometer-type (1D) compaction creep experiments performed by Liu et al. (2018), using coal powders from the same source as the present study, wet coal powder exhibits much larger compaction strains than vacuum dry coal, as well as an increase in compaction rate of 1–2 orders of magnitude. This is broadly consistent with the compaction strains estimated from the thickness change of the present wet versus vacuum dry samples, measured before versus after the experiments. This is clearly seen in Table 1 which shows 19-30% compaction strain in wet experiments S3-S8 compared with 10-19% compaction in dry samples S1 and S2. According to Liu et al. (2018), pore water enhances compaction of coal powder through a) permanent time-dependent compaction (creep) and b) the thermodynamic effect of a stress-driven reduction in water sorption capacity and an associated reduction in swelling with respect to dry material.

11. Fig. 2: I'm wondering why the background (especially for the range of 10-30 degree in two-theta) of observed intensity is so high? Is this due to a specimen holder or diffractometer used in this study?

11. Reply

The high background of observed intensity is common for coal and that is related to the amorphous component of the material (the non-aromatic component of coal in this text) and not to the specimen holder or diffractometer (e.g. Li et al. 2015; Baysal et al. 2016). In fact, we checked our measurements using a second X-ray diffractometer to test coal samples, and the background was similar. According to Franklin (1950) and other texts on XRD methods, materials with non-uniformly developed crystal structures (e.g. coal) generally show a background in the low-angel region that is high and strong. We have added this information in lines 338-343 as follows:

Fig. 8 shows that the (002), (10) and γ-side bands general characteristic of coal were observed in all samples (following Hirsch, 1954 and Lu et al., 2001). In addition, all samples showed a high background intensity, indicating that a significant proportion of amorphous carbon (i.e. non-aromatic component) was present in our coal samples (Dun et al., 2013). This high background is characteristic of materials having non-uniformly developed crystal structures (e.g. coal), regardless of the specimen holder or diffractometer used in the experiments (e.g. Li et al. 2015a;

Baysal et al. 2016).

12. Fig. 3b and 4b: I would plot apparent friction coefficients (or (a-b) values) against effective normal stress.

Fig. 6c: I can hardly see R- and Y-shear bands in the photograph. Please provide a magnified photograph.

Fig. 7: Description/citation of Fig. 7g does not appear anywhere in the main text.

12. Reply

Thanks for making us aware of this. We have revised Figs. 3b and 4b to directly show the relationship between the parameters displayed. Moreover, the highlighted region in Fig. 6c has been magnified in Fig. 6d, to clearly show the shear bands. The missing description of Fig. 7g was negligent on our part, and now we have added it in lines 332-334 as follows:

EDS data measured at the three representative spots located in Sample S4 (see Fig. 7b,c and f) are shown in Fig. 7g, indicate high C and O but little mineral content in the WDZ and PSZ zone.

13. Fig. 9: I think the horizontal axis should be an effective normal stress and/or apparent steady-state shear stress with linear scale, to indicate your data in a more scientific way. Each crystal structure parameter versus an effective normal stress and apparent steady-state shear stress should be illustrated separately using different symbol. The information of shear strain does not seem important because the values are almost identical.

Fig. 11a and 11b: Here I also suggest the horizontal axis should be an effective normal stress and/or apparent steady-state shear stress. Also please consider plotting the crystal structure parameters and Raman parameters against frictional work (shear stress*displacement, MJ/m^2 or MJ/kg ) stored in the sample, as you finally conclude that the improvement in crystallinity may be associated with strain energy.

13. Reply

Thanks for your suggestion. We have changed Fig. 9a and Fig. 11a using the apparent steady-state shear stress as a horizontal axis with a linear scale. In Fig. 9b and Fig. 11b, due to the overlap and

dispersion of data, we use the effective normal stress as a linear horizontal axis with some data offset (see the grey area in the figure). Note that in this study, we cannot capture the strain and the resulting frictional work in the shear bands upon strain localization. However, the large differences between the undeformed and deformed samples shown in Fig. 9 and Fig. 11 clearly demonstrate the role of shear deformation. To follow the reviewer's suggestion, we would need to have Raman and XRD data available at a wide range of different displacements and hence at different amounts of mechanical work (the area under the force-displacement curve up to a given displacement) applied on the sample. We fully agree that this would be a good idea, but we do not have the data needed in this study and would have to initiate another study to get it, so have decided not to plot the crystal structure and Raman parameters against the mechanical work in this study.

14. Fig. 12: Please provide "Sample number and applied stress ($\sigma$n, Pf)" in the figure legend.

14. Reply

Thanks for noting this. Now we have added it in Fig. 12.

Editor

We have received two reviews which agree on the high importance, quality, and timelines as well as clear writing of your manuscript. In agreement with the reviewers, I certainly recommend publication after addressing the comments and formatting issues, which are considered to be both minor. Two additional comments which may be helpful:

1. I agree that there is rare data available on the friction strength, stability and healing of carbonaceous materials. The authors could dig a bit on this and give a short review on or comparison with previous studies.

2. In your microstructures, you have marked many 'fractures'. If I understand correctly, you are analyzing fragments not thin section. I suggest being careful about the fractures. Some may form when you splited the sample.

1. Reply

Thanks a lot for your comments and suggestion. we have enriched the review on previous

experimental friction / shear studies of coal and related materials in Lines 73-89 and Lines 96-101 (marked in red) and we also compare our results with the literature in and Section 4.2 (lines 475-477, 488-492, 501-509), as follows:

Lines 73-101:

On the other hand, Kirilova et al. (2018) performed double direct shear experiments on dry synthetic graphitic carbon at slow slip rates of 1-100 $\mu m\ s^{-1}$ and normal stresses of 5 and 25 MPa, at room temperature. They found slip weakening of the samples from a peak frictional strength of ~0.4-0.55 to a steady-state value of ~0.15-0.25, which is higher than the steady-state $\mu$-value seen in high-velocity friction experiments on graphite. Their TEM and Raman observations suggest shear-enhanced structural disorder with increasing shear strain developing in localized slip zones. In addition, Ruan and Bhushan (1994) investigated frictional properties of highly oriented pyrolytic graphite using a friction force microscope and TEM, and found that the friction coefficient of the well-ordered carbon of (0001) plane is much smaller compared with that of the randomly-ordered carbon. These indicate internal carbon crystal structural difference may lead to a significant difference in frictional strength of graphite materials.

We now return to frictional properties of coal. O'Hara et al. (2006) performed high-velocity (1 $m\ s^{-1}$) friction experiments on high volatile bituminous coal at a normal stress of ~0.6 MPa, employing a large displacement (maximum of ~80 m). Their results demonstrated significant slip weakening behaviour and enhanced coal maturity. Specifically, the friction coefficient decreased from 0.8-1.2 to 0.1-0.4 and random vitrinite reflectance increased from ~0.6% to ~0.8%. Besides the thermal effect of shear heating, they suggested coal gasification, fluctuations in fluid pressure and gas pressurization also played a role in determining the frictional behaviour. Similar coal maturity evolution caused by frictional heating was also reported by Kitamura et al. (2012). More recent research reported by Kaneki and Hirono (2019) investigated the frictional strength of lignite, bituminous coal, anthracite and graphite by performing high-velocity (1 $m\ s^{-1}$) rotary-shear friction experiments at room temperature. They found that the peak frictional strength, for all samples, decreased with increasing maturity from 0.5 to 0.2, and the marked dynamic

weakening was observed for lignite, bituminous coal and anthracite, from peak friction coefficient values of ~0.3–0.5 to dynamic values of ~0.1–0.2. TEM, IR and Raman observations performed on the samples before and after frictional shearing suggested that the marked dynamic weakening behaviour observed in lignite, bituminous coal and anthracite was caused by a shear-induced graphitization process possibly dominated by flash heating (Kaneki and Hirono, 2019). Somewhat different results were obtained by Fan and Liu (2019). These authors performed low-velocity direct shear / friction experiments on pre-cut coal samples (low volatile bituminous coal) exposed to various fluids (helium, carbon dioxide, water, and moisturized methane) at a constant effective normal stress of 2 MPa, employing shear rates of 1–10 μm s$^{-1}$. Their results showed a) no slip-weakening, b) a steady-state friction coefficient for samples exposed to water and moisturized methane of ~0.15, c) a much higher friction coefficient in samples exposed to helium (~0.53) and carbon dioxide (~0.43), and velocity strengthening behaviour, regardless of the fluids.

Lines 473-478:

Similar slip weakening behaviour was also observed on simulated coal-shale gouges when coal content ≥50 vol% (Liu et al., 2020) and synthetic graphite gouge (Kirilova et al. 2018) under similar experimental conditions. Kirilova et al. (2018) inferred that the slip weakening behaviour seen in the synthetic graphite gouge could be related to the degree of order of crystal sheet structure in the shear zone (Morrow et al., 2000; Rutter et al., 2013). Recall that mineral content in our coal samples is ~5% only and EDS data shows little mineral content in the WDZ and PSZ zones.

Lines 488-492:

This shear-enhanced crystallinity was also observed in natural (deformed) carbonaceous materials in fault zone (Kuo et al., 2018; Wang et al., 2019), which likely results in a reduction in frictional strength (Ruan and Bhushan, 1994; Morrow et al., 2000; Moore and Lockner, 2004). Significantly, in high maturity materials such as graphite, structural disorder is enhanced by shear

deformation (Nakamura et al., 2015; Kirilova et al., 2018; Kaneki and Hirono, 2019).

Lines 501-509:

By contrast, Fan and Liu (2019) found no slip weakening behaviour in friction experiments on pre-cut bituminous coal blocks employing shear rates of 1-10 μm s$^{-1}$ to a total shear displacement of ~7 mm, at an effective normal stress of 2 MPa. This difference between the findings reported by Fan and Liu (2019) and our results suggests that strain energy may play a significant role in enhancing coal crystallinity and in reducing frictional strength. Unfortunately, we cannot directly investigate the effect of strain energy on the development of crystal structure in the present experiments, as we have insufficient experimental data. To further test the hypothesis that strain energy accumulated in the shear bands enhances crystallinity of coal, we will initiate another study involving high-resolution BIB-SEM, Raman and XRD observations on coal gouges subjected to well controlled shear displacements, such as 1, 2, 4, 5 mm and beyond, at low and high normal effective stresses.

2. Reply

Thanks for raising this point – it needs clarifying. Microfractures that are caused by sample handling can be easily recognized and we have excluded images dominated by these artificial fractures. We have added this point to lines 169-171, as follows:

Note that artificial microfractures formed during extraction of the samples from the experimental apparatus and subsequent treatment can be easily recognized and excluded.

Other modifications

We have moved the description/explanation of RSF theory from old lines 113-118 in section 2.1 to new lines 224-230 in section 2.5.1 because of redundancy in the original description. The new lines 224-230 now read as follows:

[revised manuscript text omitted]